# RSTR: Reducing SpatioTemporal Redundancy in Diffusion Transformers

Ruitong Sun [1]   Tianze Yang [1]   Wei Niu [1]   Jin Sun [1]

## Abstract

Diffusion Transformers (DiTs) have achieved remarkable success in image generation, yet their deployment is hindered by high computational costs. We identify two sources of redundancy. First, **temporal redundancy**: Classifier-Free Guidance (CFG) applies costly dual forward passes at every timestep, yet guidance matters only at specific steps, and variable scales at critical steps can compensate for skipping others. Second, **spatial redundancy**: under variable guidance, different transformer blocks exhibit heterogeneous sensitivity, yet uniform calibration across all blocks wastes computation while failing to address their varying requirements. We present RSTR, the first framework to jointly reduce spatiotemporal redundancy in diffusion transformers. Stage-1 addresses temporal redundancy through evolutionary search, discovering sparse guidance schedules with variable scales. Stage-2 addresses spatial redundancy through adaptive rank allocation, assigning calibration capacities to transformer regions based on their sensitivity. Experiments on DiT-XL/2, PixArt-$\alpha$, FLUX, and state-of-the-art Qwen-Image demonstrate 50%-70% compute savings while maintaining or improving quality. On DiT-XL/2, RSTR achieves 57% savings with 15% FID improvement; on Qwen-Image, 3.43$\times$ speedup with preserved quality.

## 1. Introduction

Diffusion models (Ho et al., 2020; Song et al., 2020b; Dhariwal & Nichol, 2021; Peebles & Xie, 2023; Chen et al., 2024a; Wu et al., 2025) have revolutionized generative modeling, achieving unprecedented quality in image synthesis and editing (Yang et al., 2025a). Yet their widespread adoption remains limited by computational demands: generating

[1]University of Georgia, Athens, GA, USA. Correspondence to: Ruitong Sun <Ruitong.Sun@uga.edu>.

*Proceedings of the 43rd International Conference on Machine Learning*, Seoul, South Korea. PMLR 306, 2026. Copyright 2026 by the author(s).

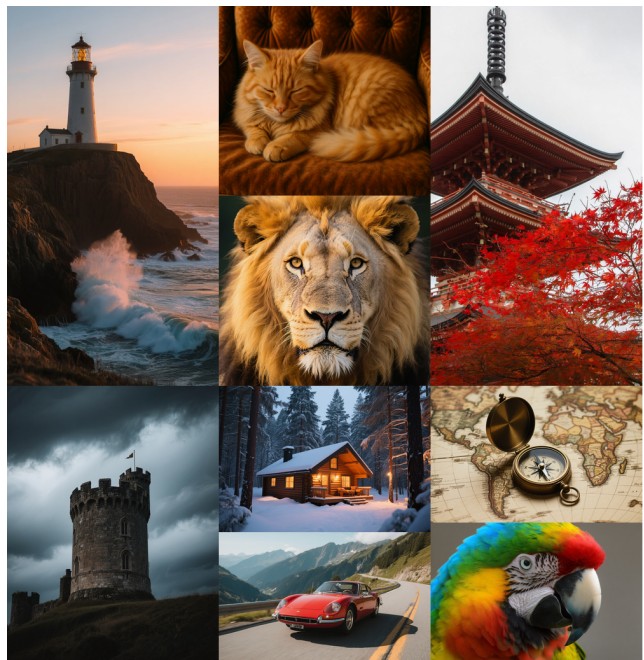

*Figure 1.* Qualitative results of RSTR on Qwen-Image at 1024$\times$1024 resolution, achieving 3.43$\times$ latency speedup while preserving generation quality across diverse text prompts.

a single high-quality image requires trillions of floating-point operations (TeraFLOPs) due to iterative denoising. This cost even doubles when using Classifier-Free Guidance (CFG) (Ho & Salimans, 2022), which improves generation quality by interpolating between conditional and unconditional predictions to strengthen adherence to input conditions. CFG remains an indispensable component of all mainstream diffusion models, yet it applies a constant guidance scale across all timesteps, ignoring that different steps have varying guidance requirements.

Existing CFG-related research pursues two distinct objectives. **Efficiency-focused methods** (Castillo et al., 2025; Yuan et al., 2024; Lyu et al., 2025) detect when conditional and unconditional outputs are similar to skip redundant computation. However, they keep guidance scales *fixed* and total sampling steps *unchanged*, achieving modest speedups while often degrading quality. **Quality-focused methods** (Gao et al., 2025a; Malarz et al., 2025a; Zhu et al., 2025;

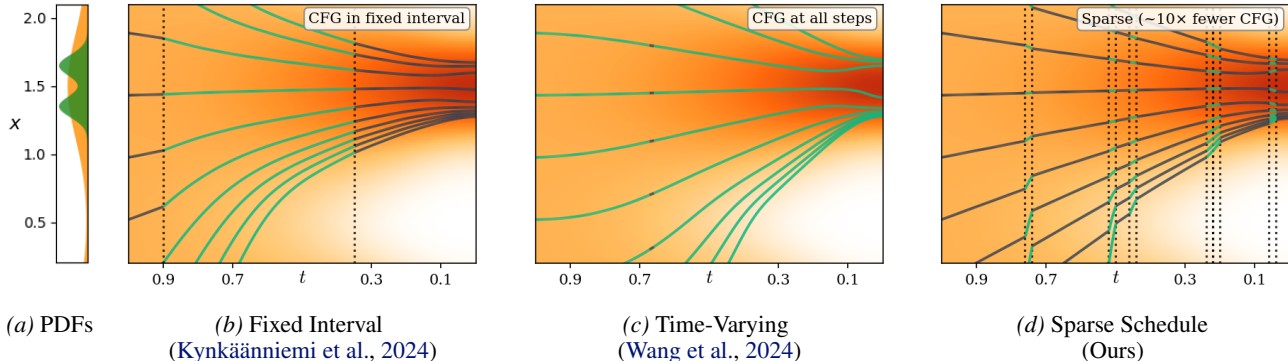

*Figure 2.* **Comparison of guidance strategies on a 1D synthetic example.** (a) Unconditional (orange) and conditional (green) distributions. (b)–(d) Sampling trajectories under different strategies, where $t$ denotes normalized noise level. All three methods reach similar final distributions, but with different computational costs and design choices: (b) Limited Interval optimizes *when* to apply guidance by restricting CFG to a fixed noise interval, but uses constant scale and requires grid search for interval boundaries; (c) Time-Varying methods optimize *what scale* to use via monotonically increasing weights, but require CFG computation at every timestep. (d) RSTR unifies both directions by jointly optimizing *when* (discrete skip patterns) and *what scale* (continuous values), discovering that guidance at only a few critical timesteps with variable scales achieves comparable quality while significantly reducing computation.

Sadat et al., 2025; Wang et al., 2024; Kynkäänniemi et al., 2024) demonstrate that non-uniform guidance strategies can improve generation quality. Yet they still require full CFG computation at every timestep, providing no computational savings, and their schedules remain "largely heuristic" (Gao et al., 2025a). These directions appear incompatible and cannot be simply combined together: one reduces computation at the cost of quality, the other improves quality without addressing efficiency.

This raises a fundamental question: *can we achieve both by jointly optimizing when to apply CFG and what guidance scale to use?* We observe that adjusting scales at critical timesteps can compensate for skipping CFG at others (Figure 2), making joint optimization both possible and beneficial. We frame this challenge as **spatiotemporal redundancy**. *Temporal redundancy* arises from applying CFG uniformly across all timesteps, yet guidance matters only at specific steps. Existing methods address only part of this problem: some restrict CFG to fixed intervals (Kynkäänniemi et al., 2024), others use time-varying scales but still require CFG at every timestep (Wang et al., 2024). Addressing temporal redundancy through variable guidance in turn reveals *spatial redundancy*: different transformer blocks exhibit varying sensitivity to guidance changes, yet existing caching methods (Chen et al., 2025) apply uniform calibration across all blocks, wasting computation on insensitive blocks while under-serving sensitive ones. These two forms of redundancy are inherently coupled and must be addressed jointly.

We present RSTR: Reducing SpatioTemporal Redundancy in Diffusion Transformers. We focus on transformers due to their growing adoption in state-of-the-art models (Peebles & Xie, 2023; Chen et al., 2024a; Labs et al., 2025)

and their uniform block structure that enables systematic optimization. RSTR employs a two-stage optimization approach. In Stage 1, we jointly optimize skip patterns and guidance scales via evolutionary search, discovering that applying variable guidance at only a few critical timesteps suffices to preserve generation quality. This poses a complex discrete-continuous optimization problem, where gradient-based methods cannot apply directly due to memory constraints and vanishing gradients over $T$ steps. However, the sparse schedules from Stage 1 introduce denoising deviations that break the feature consistency assumed by standard caching methods. Therefore, in Stage 2, we introduce adaptive rank allocation that assigns different calibration ranks to different transformer blocks based on their sensitivity to guidance variations, enabling effective feature caching under variable guidance.

Experiments on DiT-XL/2, PixArt-$\alpha$, FLUX, and state-of-the-art Qwen-Image show that RSTR achieves 50% to 70% computational savings while preserving or improving generation quality. On DiT-XL/2, RSTR achieves 15% FID improvement with 57% compute savings. On Qwen-Image, RSTR achieves 3.43× latency speedup while preserving generation quality. Our key contributions are:

- **Temporal Optimization:** We propose the first framework that jointly optimizes discrete skip patterns and continuous guidance scales via evolutionary search, reducing CFG evaluations by up to 80% while maintaining generation quality.

- **Spatial Optimization:** We introduce adaptive rank allocation that assigns different calibration capacities to different transformer regions based on their sensitivity, enabling effective caching under variable guidance where uniform-rank methods fail.

- **Insights into Guidance Dynamics:** Our analysis reveals that sparse high-scale CFG can effectively replace dense low-scale CFG, but position and scale are inherently coupled and each high-scale step is indispensable. These findings explain why learned optimization outperforms analytical derivation.

**Conflict of Interest Disclosure.** The authors declare no financial conflicts of interest related to this work.

## 2. Related Work

**Diffusion model acceleration.** Recent methods fall into three categories. Sampling acceleration reduces steps through improved solvers: DDIM (Song et al., 2020a) enables deterministic sampling, DPM-Solver (Lu et al., 2022) uses exponential integrators, and higher-order methods (Zhang & Chen, 2022) further improve convergence. Distillation (Salimans & Ho, 2022; Park et al., 2025) reduces steps but requires training or cannot generalize across guidance scales. Architectural optimizations reduce per-step costs through efficient designs (Peebles & Xie, 2023), pruning (Zhu et al., 2024), and quantization (Liu et al., 2025c). Token-level analysis (Yang et al., 2025b) further reveals which discrete representations are critical for generation. The need for acceleration extends to health-related diffusion (Li et al., 2025c) and medical imaging more broadly (Sun & Rostami, 2024; 2026). Our work maintains full denoising steps while reducing per-step cost through selective CFG forward passes and adaptive caching, without architectural modifications.

**Dynamic guidance scheduling.** Evidence shows constant CFG wastes computation. Kynkäänniemi et al. (2024) find guidance harmful at extreme noise levels, while Wang et al. (2024) shows monotonic schedules outperform constant guidance. Theoretical advances include progressive guidance (WANG et al., 2024), characteristic guidance with nonlinear corrections (Zheng & Lan, 2024), and gradient artifact correction (Gao et al., 2025b). Methods for reducing CFG cost include convergence detection (Castillo et al., 2025), early-stage compression (Dinh et al., 2024), and adaptive scaling (Malarz et al., 2025b; Li et al., 2025b). Other works explore time-varying scales for quality improvement (Gao et al., 2025a; Malarz et al., 2025a; Zhu et al., 2025; Sadat et al., 2025). Zhang & Wan (2025) and Yehezkel et al. (2025) showed optimal schedules vary across architectures. Alternative approaches include autoguidance (Karras et al., 2024) and condition annealing (Sadat et al., 2024). We are the first to use evolutionary optimization to jointly search discrete skip patterns and continuous guidance scales.

**Feature caching and calibration.** Caching exploits temporal redundancy between timesteps. DeepCache (Ma et al., 2024b) pioneered feature reuse for U-Nets, while

TGATE (Liu et al., 2024) enables selective attention caching. Extensions to transformers include block-level caching (Wimbauer et al., 2024), training-inference harmonization (Huang et al., 2025), and stage-aware block caching in Δ-DiT (Chen et al., 2024b). Learning-to-Cache (Ma et al., 2024a) uses learned routing, and token-wise caching (Zou et al., 2024) achieves 2.36× speedup through selective token reuse. TeaCache (Liu et al., 2025a) leverages timestep embeddings for adaptive caching, TaylorSeer (Liu et al., 2025b) predicts features via Taylor expansion, and ICC (Chen et al., 2025) combines caching with uniform SVD calibration. The notion of calibration across network layers has also been examined in large language models (Joshi et al., 2025), alongside related advances in reasoning (Li et al., 2025a) and multimodal alignment (Liu et al., 2026). However, no existing work addresses how calibration should adapt when guidance varies.

## 3. Preliminaries

**Classifier-free guidance.** To improve the quality of conditional generation, CFG (Ho & Salimans, 2022) interpolates between conditional and unconditional predictions:

$$\tilde{\epsilon}_\theta(x_t, c, t) = \epsilon_\theta(x_t, \emptyset, t) + w \cdot (\epsilon_\theta(x_t, c, t) - \epsilon_\theta(x_t, \emptyset, t)), \tag{1}$$

where $c$ denotes the conditioning information, $\emptyset$ represents null conditioning, and $w$ controls the guidance scale. Applying CFG at each timestep *doubled* the total computation.

**Incremental calibration.** Recent work (Chen et al., 2025) accelerates diffusion transformers by caching and reusing features across timesteps. The method corrects cached features through layer-wise calibration:

$$\hat{\mathbf{h}}_{\text{out}}^\ell = \mathcal{P}(\mathbf{h}_{\text{out}}^{\ell,\text{prev}}) + \mathbf{A}^\ell(\mathbf{h}_{\text{in}}^\ell - \mathcal{P}(\mathbf{h}_{\text{in}}^{\ell,\text{prev}})), \tag{2}$$

where $\ell$ is the layer index, $\mathcal{P}(\cdot)$ denotes the caching operation from the previous timestep, $\mathbf{h}_{\text{in}}^\ell$ is the current input to layer $\ell$, and $\mathcal{P}(\mathbf{h}_{\text{in}}^{\ell,\text{prev}})$ and $\mathcal{P}(\mathbf{h}_{\text{out}}^{\ell,\text{prev}})$ are the cached input and output from the previous timestep. Each layer has its own calibration matrix $\mathbf{A}^\ell$ that transforms the input increment to correct the cached output. To reduce computation, each $\mathbf{A}^\ell$ is approximated via SVD: $\mathbf{A}^\ell = \mathbf{U}^\ell \mathbf{\Sigma}^\ell \mathbf{V}^{\ell T} \approx \mathbf{U}_r^\ell \mathbf{\Sigma}_r^\ell \mathbf{V}_r^{\ell T}$, where subscript $r$ denotes truncation to rank $r$. Prior increment-calibrated caching methods use uniform rank $r$ across all transformer blocks.

## 4. RSTR

### 4.1. Overview

Standard diffusion inference suffers from two forms of redundancy. First, CFG applies dual forward passes at every timestep, yet guidance matters only at specific steps. This *temporal redundancy* suggests that many unconditional

STAGE 1: Discovering sparse guidance schedules                    STAGE 2: Adaptive rank allocation

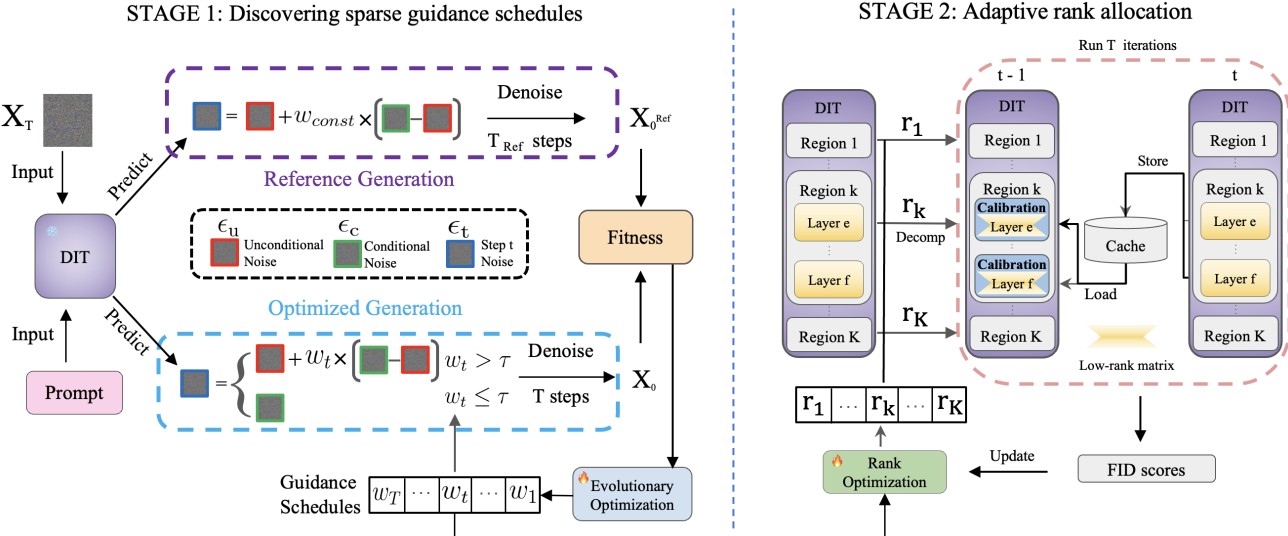

*Figure 3.* **The two-stage RSTR optimization framework. Stage 1 (Left):** Evolutionary optimization discovers sparse guidance schedules by refining per-step guidance $\mathbf{w} = [w_T, \ldots, w_1]$. Starting from noise $x_T$, the framework generates a reference $x_0^{\text{Ref}}$ via $T_{\text{Ref}}$ denoising steps. At each timestep, full CFG is applied if $w_t > \tau$, otherwise only conditional forward passes are performed. The fitness function balances quality and sparsity, iteratively improving $\mathbf{w}$ through population sampling and evaluation. **Stage 2 (Right):** Adaptive rank allocation optimizes caching. DiT blocks are partitioned into $K$ regions, each assigned a calibration rank $r_k$. Cached features are corrected via SVD-based calibration. Coordinate descent with binary search tunes ranks to minimize FID.

passes can be eliminated. However, even with sparse guidance, conditional passes remain at every timestep. To further reduce computation, we turn to feature caching, but variable guidance reveals *spatial redundancy*: different transformer blocks require different calibration capacities. We address these two forms of redundancy as follows: Section 4.2 tackles temporal redundancy through evolutionary search for sparse guidance schedules; Section 4.3 analyzes why variable guidance challenges standard caching; Section 4.4 tackles spatial redundancy through adaptive rank allocation.

### 4.2. Stage 1: Discovering Sparse Guidance Schedules

CFG's computational cost comes from applying guidance uniformly at every denoising timestep. This *temporal redundancy* arises because guidance matters only at specific steps, yet standard CFG performs dual forward passes at all steps. Prior work (Kynkäänniemi et al., 2024) shows that guidance can be harmful at extreme noise levels and unnecessary near convergence, motivating interval-based strategies. We go further and explore whether non-contiguous, task-specific patterns can outperform fixed intervals. We start by replacing the constant guidance scale $w$ with a per-timestep schedule to reformulate Equation 1:

$$\tilde{\epsilon}_\theta = \epsilon_\theta(x_t, \emptyset, t) + w_t \cdot [\epsilon_\theta(x_t, c, t) - \epsilon_\theta(x_t, \emptyset, t)], \quad (3)$$

where $w_t$ is now timestep-dependent. We optimize a guidance schedule $\mathbf{w} = [w_1, \ldots, w_T]$ where each $w_t \in [1, w_{\max}]$. When $w_t$ falls below a threshold $\tau$, we set $w_t = 1$ and skip the unconditional forward pass, performing only

the conditional pass.

The best schedule $\mathbf{w}$ can be found by solving the following optimization problem:

$$\mathbf{w}^* = \arg\min_{\mathbf{w}} \mathcal{L}_{\text{total}}(\mathbf{w}) = \mathcal{L}_{\text{quality}}(\mathbf{w}) + \lambda \mathcal{L}_{\text{sparse}}(\mathbf{w}) \quad (4)$$

The quality preservation term $\mathcal{L}_{\text{quality}}$ ensures sparse schedules maintain generation fidelity through output matching:

$$\mathcal{L}_{\text{quality}}(\mathbf{w}) = \mathbb{E}_{x_T, c} \left[ \| \mathcal{G}_T(x_T, c; \mathbf{w}) - \mathcal{G}_{T_{\text{ref}}}(x_T, c; w_{\text{const}}) \|_2^2 \right] \quad (5)$$

where $\mathcal{G}_T$ denotes the $T$-step generation process starting from initial noise $x_T$ with our optimized schedule $\mathbf{w}$, and $\mathcal{G}_{T_{\text{ref}}}$ represents reference generation from the same $x_T$ using constant guidance $w_{\text{const}}$. The reference uses substantially more steps ($T_{\text{ref}} \gg T$, typically 100-1000 vs 20-50) to provide smooth denoising process and high-quality targets. Starting from identical noise ensures fair comparison and helps identify critical timesteps for guidance.

The sparsity term directly penalizes the number of timesteps requiring full CFG forward pass: $\mathcal{L}_{\text{sparse}}(\mathbf{w}) = \sum_{t=1}^{T} \mathbb{I}[w_t > \tau]$ where $\tau$ serves as an activation threshold below which guidance is completely disabled, eliminating the unconditional forward pass. This binary decision at each timestep transforms the optimization into a hybrid continuous-discrete problem (Barton et al., 2000).

**Evolutionary Optimization Strategy** Direct gradient-based optimization of this objective is intractable as it would require backpropagation through the entire $T$-step generation trajectory, creating prohibitive memory requirements and suffering from vanishing gradients. Instead, we employ a tailored evolutionary strategy that operates in a transformed space for numerical stability.

We maintain a population center $\boldsymbol{\mu} \in \mathbb{R}^T$ where $\boldsymbol{\mu} = [\mu_1, \ldots, \mu_T]^T$ with each $\mu_t \in \mathbb{R}$. This center represents the mean of our search distribution in the parameter space, a fundamental concept in both CMA-ES (Hansen, 2023) and Natural Evolution Strategies (Wierstra et al., 2014; Yi et al., 2009). At each generation $g \in \{1, \ldots, G\}$, we decode the center to get base guidance values $\mathbf{w}_{\text{base}} = w_{\max} \cdot \text{sigmoid}(\boldsymbol{\mu_g})$. We construct a population by perturbing these base values. For each candidate $i \in \{1, \ldots, P\}$: $\mathbf{w}^{(i)} = \mathbf{w}_{\text{base}} + \boldsymbol{\delta}^{(i)}$, where $\boldsymbol{\delta}^{(i)} \sim \mathcal{N}(0, \sigma_{\text{noise}}^2 I)$ and $\sigma_{\text{noise}} = \sigma_0(1 - g/G)$ decreases across generations to refine exploration, with $\sigma_0$ being the initial noise scale.

We apply a threshold $\tau$ to sparsify the guidance schedule and determine the noise prediction at $t$:

$$\epsilon_t = \begin{cases} \epsilon_u + w_t^{(i)} \cdot (\epsilon_c - \epsilon_u) & \text{if } w_t^{(i)} \geq \tau \\ \epsilon_c & \text{if } w_t^{(i)} < \tau \end{cases} \quad (6)$$

Each candidate's fitness is computed as $f^{(i)} = -\mathcal{L}_{\text{quality}}(\mathbf{w}^{(i)}) + \lambda \cdot S(\mathbf{w}^{(i)})$, where $S(\mathbf{w}^{(i)}) = (T - \|\mathbf{w}^{(i)}\|_0)/T$ measures sparsity. Given fitness values $\{f^{(1)}, \cdots, f^{(P)}\}$, we compute rank-based weights $a_i = d_i/(P-1) - 0.5$, where $d_i \in \{0, 1, \cdots, P-1\}$ is the rank of candidate $i$, with 0 for the lowest fitness and $P-1$ for the highest. This rank-based weighting scheme follows established practices in evolution strategies (Hansen & Ostermeier, 2001).

The population center evolves through natural gradient estimation: $\boldsymbol{\mu_{g+1}} \leftarrow \boldsymbol{\mu_g} + \frac{\eta}{P} \sum_{i=1}^{P} a_i \cdot (\text{sigmoid}^{-1}(\mathbf{w}^{(i)}/w_{\max}) - \boldsymbol{\mu_g})$. After $G$ generations, the converged center $\boldsymbol{\mu}^*$ yields $\mathbf{w}^* = w_{\max} \cdot \text{sigmoid}(\boldsymbol{\mu}^*)$. This sparse schedule $\mathbf{w}^*$ applies guidance selectively at critical timesteps to reduce redundant computations. We provide theoretical justification for this search space design in Appendix A.

**From Sparse Guidance to Efficient Execution** Stage 1 achieves significant savings by eliminating unconditional forward passes at non-critical timesteps. However, the conditional forward passes at all timesteps still dominate the remaining computational cost. To fully realize the efficiency gains, we reduce this redundancy through feature caching. Yet as we show next, our variable guidance schedules from Stage 1 fundamentally challenge standard caching methods.

## 4.3. The Challenge of Caching with Variable Guidance

The sparse guidance schedules discovered in *Stage 1* fundamentally challenge existing caching methods. Standard caching approaches such as ICC (Chen et al., 2025) assume that features between consecutive timesteps remain similar. This assumption holds when guidance remains constant throughout denoising. However, our variable guidance patterns from *Stage 1* break this assumption, causing the incremental calibration in Equation 2 to become less effective when correcting the larger feature differences introduced by changing guidance scales.

As shown in Figure 6, when we apply variable guidance from *Stage 1* with naive caching, feature reconstruction errors increase significantly compared to constant CFG with the same caching method. The variable guidance causes higher MSE across all transformer blocks, with particularly severe degradation in deeper blocks. Since these elevated reconstruction errors accumulate through the network and degrade final image quality, which motivates us to explore the reason behind this.

**Root Cause Analysis.** To understand the source of these elevated reconstruction errors, we analyze the denoising deviations introduced by variable guidance. We identify two primary scenarios that exceed standard calibration capacity. We consider two scenarios: when consecutive timesteps both use CFG but with different guidance scales, and when timesteps switch between using CFG and using only conditional prediction. Each scenario introduces distinct types of denoising fluctuations that degrade caching.

*Scenario 1: Both timesteps use CFG with different scales.* When both $w_t^* > \tau$ and $w_{t-1}^* > \tau$, the denoising process deviates by:

$$\Delta x_{t-1}^{\text{strength}} = \sqrt{\bar{\alpha}_{t-1}/\bar{\alpha}_t} \cdot (w_{t-1}^* - w_t^*) \cdot [\epsilon_c(x_t, t) - \epsilon_u(x_t, t)] \quad (7)$$

where $\bar{\alpha}_t$ is the cumulative noise schedule coefficient. This deviation grows with the guidance difference $(w_{t-1}^* - w_t^*)$ and the gap between conditional and unconditional predictions.

*Scenario 2: CFG at timestep $t$ transitions to no guidance at timestep $t - 1$.* When $t$ uses CFG ($w_t^* > \tau$) but $t - 1$ uses only conditional prediction ($w_{t-1}^* < \tau$), the deviation becomes:

$$\Delta x_{t-1}^{\text{switch}} = \beta_{t-1,t} \cdot (1 - w_t^*) \cdot [\epsilon_c(x_t, t) - \epsilon_u(x_t, t)], \quad (8)$$

where $\beta_{t-1,t} = \sqrt{1 - \bar{\alpha}_{t-1}} - \sqrt{\bar{\alpha}_{t-1}(1 - \bar{\alpha}_t)/\bar{\alpha}_t}$ is the noise coefficient for DDIM sampling. Detailed mathematical derivations are provided in Appendix B.1. These deviations exceed what standard caching methods can handle, as cached features from timestep $t$ no longer match the expected input for timestep $t - 1$. This mismatch causes

higher reconstruction errors that accumulate through the transformer blocks. This heterogeneous error distribution reveals that different transformer regions require different calibration to effectively handle variable guidance.

### 4.4. Stage 2: Adaptive Rank Allocation

While incremental calibration can mitigate caching errors, a critical question remains: should all transformer blocks use the same calibration rank? Uniform calibration ignores potential differences in block sensitivity, resulting in *spatial redundancy*. We now examine this through experiments.

Caching with variable guidance causes larger denoising deviations than caching under a constant CFG. While incremental calibration was designed to alleviate caching errors under constant CFG, we now examine how it performs under variable guidance patterns. In our experiment, we find that different blocks benefit from different calibration ranks. Specifically, the comparison between $r = 256$ and $r = 512$ reveals a counter-intuitive phenomenon: while the higher rank $r = 512$ effectively suppresses severe error spikes in Block 22, it introduces higher errors in Blocks 0, 14, and 26 compared to $r = 256$. This heterogeneous pattern demonstrates that no single uniform rank is optimal for all blocks. Although high-capacity calibration ($r = 1024$) consistently minimizes errors across all blocks as shown in Figure 8, it incurs prohibitive computational cost. Much of this cost is spatial redundancy, as many blocks require far less calibration capacity. For detailed analysis across all transformer blocks, please refer to Appendix B.2. These observations motivate our systematic approach to discovering optimal rank distributions, where we assign each region the rank that best balances error reduction and efficiency.

Formally, we partition the $N$ transformer blocks into $K$ regions $\mathcal{R} = \{R_1, \ldots, R_K\}$ based on their network position. We divide blocks uniformly such that each region contains $\lfloor N/K \rfloor$ consecutive blocks, with region $R_k$ containing blocks from index $(k-1) \cdot \lfloor N/K \rfloor$ to $k \cdot \lfloor N/K \rfloor - 1$. Each region $k$ receives a tailored calibration rank $r_k$, yielding a region-specific calibration matrix $\mathbf{A}_\ell \approx \mathbf{U}_{\ell, r_k} \mathbf{\Sigma}_{\ell, r_k} \mathbf{V}_{\ell, r_k}^T$ for layer $\ell \in R_k$. We optimize the rank configuration $\mathbf{r} = [r_1, r_2, \ldots, r_K]$ where each $r_k \in [r_{\min}, r_{\max}]$.

**Rank Optimization via Coordinate Descent.** Finding the optimal rank configuration $\mathbf{r}^* = [r_1, \ldots, r_K]$ requires searching over a large discrete space where each region can take ranks from $[r_{\min}, r_{\max}]$. This is challenging because rank assignments across regions interact through the sequential nature of the transformer: early blocks affect later blocks' inputs, creating complex dependencies that make the relationship between rank configuration and generation quality non-linear. We formulate this as minimizing $\mathbf{r}^* = \arg \min_{\mathbf{r}} \text{FID}(\mathcal{G}_T(\mathbf{r}, \mathbf{w}^*))$ subject to $r_k \in [r_{\min}, r_{\max}]$ and

a total budget constraint $\sum_{k=1}^{K} r_k \leq B$, where $\mathcal{G}_T(\mathbf{r}, \mathbf{w}^*)$ denotes the $T$-step generation process using rank configuration $\mathbf{r}$ with the optimized guidance schedule $\mathbf{w}^*$ from *Stage 1*. We solve this optimization through coordinate descent (Wright, 2015), decomposing the problem into a sequence of single-variable optimizations. For each region $k$, we fix the ranks of all other regions and search for $r_k^* = \arg \min_{r_k} \text{FID}(\mathcal{G}(r_1, \ldots, r_k, \ldots, r_K, \mathbf{w}^*))$. Within each coordinate optimization step, we use binary search to efficiently explore the rank space $[r_{\min}, r_{\max}]$. This procedure iterates across all regions until the overall rank configuration converges.

## 5. Experiments

### 5.1. Experimental Setup

**Models and datasets.** We evaluate RSTR on four diffusion transformers: DiT-XL/2 (Peebles & Xie, 2023) on ImageNet (Deng et al., 2009) (50K images, $256^2$ and $512^2$); PixArt-$\alpha$ (Chen et al., 2024a) on MSCOCO 2014 (Lin et al., 2014) (30K images, $256^2$) using captions from (Zou et al., 2024); FLUX (Labs et al., 2025) with True CFG on DrawBench (Saharia et al., 2022) and GenEval (Ghosh et al., 2023) ($512^2$); and Qwen-Image (Wu et al., 2025) on GenEval, GenEval2, and DrawBench ($1024^2$).

**Evaluation metrics.** For ImageNet, we report IS (Salimans et al., 2016), FID (Nash et al., 2021), sFID, and Precision-Recall (Kynkäänniemi et al., 2019). For MSCOCO, we report FID-30k, CLIP Score (Hessel et al., 2021), and CLIP Score on PartiPrompts (Yu et al., 2022). For FLUX and Qwen-Image, we report ImageReward (Xu et al., 2023), CLIP Score on DrawBench (Saharia et al., 2022), and compositional metrics on GenEval (Ghosh et al., 2023). For Qwen-Image, we additionally report Soft-TIFA AM and GM on GenEval2 (Kamath et al., 2025). Efficiency is measured via MACs; latency uses batch size 8 on a single H100.

**Baselines.** We compare against: ICC (Chen et al., 2025) (uniform-rank caching), L2C (Ma et al., 2024a) (learned routing), HarmoniCa (Huang et al., 2025) (training-inference harmonization), TaylorSeer (Liu et al., 2025b) (Taylor expansion prediction), Adaptive Guidance (Castillo et al., 2025) (similarity-based CFG skipping), and DDIM/DPM-Solver at various step counts.

### 5.2. Main Results

**DiT-XL/2 on ImageNet.** Table 1 demonstrates substantial efficiency gains across both resolutions. At $512 \times 512$, RSTR applies guidance at only 9 of 50 timesteps, matching 50-step DDIM quality (FID 2.72 vs 3.20) with 47% less computation (24.97T vs 52.45T MACs) and even surpassing 1000-step DDIM (FID 2.72 vs 2.99). At $256 \times 256$, RSTR achieves the best FID (2.04) among all baselines, including

*Table 1.* DiT-XL/2 results on ImageNet. Best in bold. †: without adaptive caching.

| Methods | $T$ | $T_{cfg}$ ↓ | MACs↓ | Lat.↓ | IS↑ | FID↓ | sFID↓ | Pr.↑ | Re.↑ |
|---|---|---|---|---|---|---|---|---|---|
| *DiT-XL/2 (ImageNet 512×512)* | | | | | | | | | |
| DDIM | 1000 | 1000 | 1049 | 416.8 | 210.6 | 2.99 | 4.38 | 83.2 | 55.6 |
| DDIM | 50 | 50 | 52.5 | 20.86 | 203.8 | 3.20 | 4.53 | 83.3 | 56.4 |
| L2C | 50 | 50 | 40.6 | 16.30 | 199.5 | 3.98 | 5.66 | 82.5 | 53.3 |
| ICC | 50 | 50 | 33.4 | 15.88 | 200.0 | 3.73 | 5.39 | 83.3 | 55.6 |
| TaylorSeer | 50 | 50 | 18.9 | 12.79 | 201.2 | 3.51 | 4.37 | 83.5 | 53.3 |
| RSTR† | 50 | 9 | 30.9 | 12.91 | 228.3 | **2.71** | 4.38 | 83.4 | **57.0** |
| RSTR | 50 | 9 | **24.9** | **11.28** | 228.6 | 2.72 | 4.13 | 83.6 | 55.3 |
| DDIM | 30 | 30 | 31.5 | 12.52 | 198.0 | 3.86 | 4.94 | **83.1** | 54.4 |
| L2C | 30 | 30 | 25.7 | 10.31 | 189.4 | 4.93 | 6.72 | 82.1 | 54.5 |
| ICC | 30 | 30 | 20.1 | 9.54 | 171.0 | 6.85 | 6.72 | 79.8 | 53.5 |
| RSTR† | 30 | 8 | 19.9 | 8.20 | 214.8 | 3.33 | 4.85 | 83.0 | 55.3 |
| RSTR | 30 | 8 | **16.0** | **7.19** | 209.7 | 3.37 | 4.66 | 82.6 | **56.5** |
| *DiT-XL/2 (ImageNet 256×256)* | | | | | | | | | |
| DDIM | 1000 | 1000 | 237 | 106.4 | 245.0 | 2.12 | 4.66 | 80.7 | 59.7 |
| DDIM | 50 | 50 | 11.9 | 5.33 | 239.4 | 2.23 | 4.29 | 80.1 | 59.2 |
| HarmoniCa | 50 | 50 | 10.6 | 4.78 | 210.1 | 3.33 | 5.03 | 77.4 | **60.1** |
| L2C | 50 | 50 | 9.8 | 4.43 | 245.5 | 2.23 | **4.27** | 81.0 | 59.1 |
| ICC | 50 | 50 | 8.1 | 4.18 | 258.5 | 2.16 | 4.28 | 82.1 | 58.1 |
| RSTR† | 50 | 8 | 6.6 | 3.37 | 263.0 | 2.10 | 4.29 | 81.9 | 58.8 |
| RSTR | 50 | 8 | **5.1** | **2.81** | 266.5 | 2.04 | 4.29 | 82.1 | 58.7 |

*Table 2.* PixArt-$\alpha$ results on MSCOCO (256×256). †: without adaptive caching.

| Method | $T$ | $T_{cfg}$ ↓ | MACs↓ | FID↓ | CLIP$_{COCO}$↑ | CLIP$_{Parti}$↑ |
|---|---|---|---|---|---|---|
| DPM-Solver | 1000 | 1000 | 336.11 | 22.97 | 16.42 | 17.31 |
| DPM-Solver | 20 | 20 | 6.72 | 24.60 | 16.31 | 17.36 |
| ICC | 20 | 20 | 3.70 | 21.86 | 16.47 | 17.20 |
| RSTR† | 20 | 6 | 4.37 | 22.69 | 16.39 | **17.92** |
| RSTR | 20 | 6 | **2.67** | **19.27** | 16.48 | 17.45 |

1000-step DDIM (2.12).

*Table 3.* FLUX results on DrawBench and GenEval at $512 \times 512$ resolution. IR: ImageReward. †: without adaptive caching.

| Method | $T$ | $T_{cfg}$ ↓ | Lat.↓ | MACs↓ | IR↑ | CLIP↑ | GenEval↑ |
|---|---|---|---|---|---|---|---|
| Origin | 50 | 50 | 52.01 | 1143.82 | 1.0074 | 27.79 | 68.60 |
| Origin | 20 | 20 | 21.24 | 457.52 | 0.8950 | 27.55 | 66.05 |
| Origin | 16 | 16 | 17.19 | 366.02 | 0.7866 | 27.46 | 63.59 |
| Origin | 50 | × | 26.43 | 571.48 | 0.9296 | 27.23 | 65.38 |
| Origin | 20 | × | 10.87 | 228.76 | 0.8934 | 27.03 | 65.26 |
| ICC$^{N=6}$ | 50 | 50 | 52.45 | 662.06 | 0.8135 | 27.64 | 62.19 |
| Ada Guidance | 20 | 16 | 19.62 | 411.77 | 0.8928 | 27.55 | 65.84 |
| TaylorSeer$^{N=3}$ | 50 | 50 | 28.30 | 411.77 | 1.0022 | 27.78 | **67.70** |
| RSTR† | 20 | 8 | 15.15 | 320.26 | **1.0092** | **28.06** | 67.46 |
| TaylorSeer$^{N=6}$ | 50 | 50 | 20.10 | 228.76 | 0.8492 | 27.26 | 59.58 |
| TaylorSeer$^{N=3}$ | 50 | × | 13.87 | 205.88 | 0.9445 | 27.37 | 65.93 |
| RSTR | 20 | 8 | 14.72 | 190.36 | **0.9620** | 28.05 | **66.47** |

**PixArt-$\alpha$ on MSCOCO.** Table 2 shows that RSTR discovers only 6 of 20 timesteps need guidance, achieving FID 19.27 (21.7% improvement) with 60% computational reduction (2.67 vs 6.72 MACs) while maintaining text-image alignment (CLIP score 16.48 vs 16.31).

*Table 4.* Qwen-Image results on GenEval, GenEval2, and Draw-Bench at $1024 \times 1024$ resolution. †: without adaptive caching.

| Method | $T$ | $T_{cfg}$ ↓ | MACs↓ | Lat.↓ | GenEval↑ | GenEval2↑ | | DrawBench | |
|---|---|---|---|---|---|---|---|---|---|
| | | | | | | AM | GM | IR↑ | CLIP↑ |
| Origin | 50 | 50 | 4183.72 | 56.73 | 88.40 | 81.56 | 38.72 | 1.206 | 29.13 |
| Origin | 15 | 15 | 1255.11 | 17.56 | 87.79 | 80.98 | 39.55 | 1.132 | 28.90 |
| Origin | 10 | 10 | 836.74 | 12.12 | 86.53 | 78.92 | 37.07 | 1.008 | 28.73 |
| ICC$^{N=6}$ | 50 | 50 | 2410.01 | 35.47 | 85.39 | 79.15 | 36.77 | 1.085 | 28.97 |
| TaylorSeer$^{N=3}$ | 50 | 50 | 1589.81 | 36.72 | 87.54 | 81.22 | 38.76 | **1.195** | **29.05** |
| Ada Guidance | 20 | 11 | 1296.95 | 17.10 | 87.85 | **81.27** | 38.97 | 1.152 | 28.83 |
| RSTR† | 20 | 10 | 1255.11 | 16.53 | **88.11** | 80.87 | **40.41** | 1.184 | 28.90 |
| TaylorSeer$^{N=6}$ | 50 | 50 | 920.41 | 29.27 | 79.47 | 78.03 | 37.09 | 0.832 | 27.77 |
| RSTR | 20 | 10 | 885.20 | 16.62 | 87.21 | 80.81 | 40.57 | 1.172 | 28.92 |

**FLUX on DrawBench and GenEval.** Table 3 evaluates RSTR on FLUX with True CFG. RSTR discovers that only 8 of 20 timesteps require guidance. On DrawBench, RSTR without caching achieves the highest ImageReward (1.0092) and CLIP Score (28.06), outperforming the 50-step full CFG baseline with 72% less computation. On GenEval, RSTR achieves an overall score of 67.46, approaching the 68.60 of 50-step full CFG.

**Qwen-Image on GenEval, GenEval2, and DrawBench.** Table 4 evaluates RSTR on Qwen-Image. RSTR discovers that only 10 of 20 timesteps require guidance, achieving $4.7\times$ speedup (56.73s → 16.62s) while maintaining quality on GenEval (87.21 vs 88.40) and improving GenEval2 GM (40.57 vs 38.72).

### 5.3. Analysis

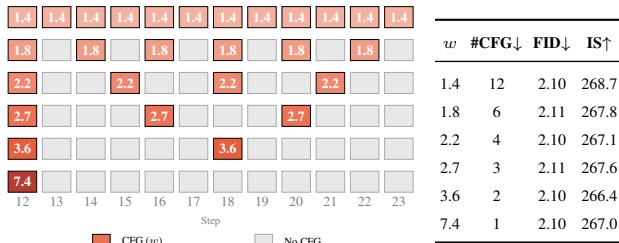

*Figure 4.* Guidance concentration principle. Within a 12-step interval (steps 12–23), sparse high-scale CFG achieves comparable quality to dense low-scale CFG with significantly fewer evaluations (12× reduction).

**Verification of Learned Guidance Schedule.** To understand why sparse schedules work, we analyze the Stage 1 schedule discovered on DiT-XL/2 256×256 with 50 steps. The learned schedule applies $w$=7.4 at step 12, with the following 11 steps (steps 13–23) requiring no CFG—the longest consecutive no-CFG interval, suggesting that *a single high-scale guidance step suffices to satisfy the guidance needs of an entire interval*.

To validate, we replace this 12-step interval with varying densities of low-scale CFG (Figure 4). Dense guidance ($w$=1.4, 12 evaluations) achieves FID 2.10. Reducing density while increasing scale maintains comparable FID: every two steps ($w$=1.8, 6 evaluations), every three steps ($w$=2.2, 4 evaluations), down to step 12 alone ($w$=7.4, 1 evaluation). This confirms a $12\times$ CFG reduction within this interval.

*Table 5.* CFG position sensitivity.

| Position | IS↑ | FID↓ |
|---|---|---|
| step 8 | 256.5 | 2.04 |
| step 10 | 260.2 | 2.08 |
| step 12 (orig.) | 263.0 | 2.10 |
| step 14 | 268.0 | 2.18 |
| step 16 | 272.4 | 2.21 |

Crucially, position and scale are coupled. Shifting $w$=7.4 from step 12 to other positions (Table 5) reveals: moving later (step 16) improves IS but worsens FID, as the same scale becomes excessive for the shorter 8-step coverage; moving earlier (step 8) decreases IS but improves FID, as the scale becomes insufficient for the longer 16-step coverage. Meanwhile, removing any high-scale step (Table 6) degrades both IS and FID, indicating that coverage regions do not overlap sufficiently to compensate for each other.

*Table 6.* Ablation on high-scale CFG steps.

| Removed | IS↑ | FID↓ |
|---|---|---|
| None | 263.0 | 2.10 |
| step 24 | 233.2 | 2.26 |
| step 27 | 237.9 | 2.25 |
| step 38 | 240.4 | 2.30 |
| step 39 | 230.8 | 2.44 |
| step 47 | 255.9 | 2.16 |

These results demonstrate that: (1) the relationship between guidance scale and effective range is non-linear, (2) position and scale cannot be adjusted independently, and (3) each high-scale step is indispensable. This explains why Stage 1 jointly optimizes both dimensions through learned search rather than analytical derivation.

*Table 7.* Guidance schedule transferability (DiT-XL/2, 256×256). Scaling factors $k$ applied to $\mathbf{w}^*$. Equivalent constant CFG scales are 1.5, 5.0, 7.5 for $1.0\times$, $3.3\times$, $5.0\times$.

| | | | FID↓ | | | sFID↓ | | |
|---|---|---|---|---|---|---|---|---|
| Method | T | $\mathbf{T_{cfg}}$↓ | 1.0× | 3.3× | 5.0× | 1.0× | 3.3× | 5.0× |
| DDIM | 50 | 50 | 2.23 | 16.39 | 19.71 | 4.29 | 15.53 | 19.95 |
| RSTR† | 50 | 8 | **2.10** | **9.20** | **11.76** | 4.29 | **9.23** | **13.20** |

**Guidance Schedule Transferability**  Table 7 evaluates whether discovered schedules generalize beyond their calibration point by applying multiplicative scaling factors $k$ to $\mathbf{w}^*$. Scaling by $3.3\times$ causes severe degradation for constant CFG (FID 2.23 $\rightarrow$ 16.39) while RSTR degrades gracefully (2.10 $\rightarrow$ 9.20). This confirms that sparse schedules effectively filter harmful guidance signals while retaining beneficial ones. Additional transferability results on FLUX and Qwen-Image are provided in Appendix E.3.

**Cross-Model-Size Transfer**  We test whether a schedule optimized on one model transfers across *model sizes* within the same family: we apply the DiT-XL/2 (675M) schedule, without re-optimization, to the smaller DiT-B/2 (131M) and

DiT-S/2 (33M) using the released T-Stitch checkpoints (Pan et al., 2025), with guidance at only 8 of 50 timesteps. Since the three models are trained for different iteration counts, absolute FID is not comparable across sizes, so we report the improvement *within* each size (Table 8). The transferred schedule improves FID over the constant-CFG baseline at both sizes—DiT-B/2 11.52$\rightarrow$**9.22** ($-20\%$) and DiT-S/2 24.09$\rightarrow$**20.00** ($-17\%$)—with consistent sFID and Precision gains, indicating that it captures intrinsic, timestep-level guidance importance shared across DiT variants of different sizes rather than artifacts of the model it was optimized on.

*Table 8.* Cross-model-size transfer: the RSTR† schedule optimized on DiT-XL/2 is applied to DiT-B/2 and DiT-S/2 without re-optimization (ImageNet $256 \times 256$, DDIM 50 steps, guidance at 8/50; source DiT-XL/2: FID 2.23$\rightarrow$2.10). Absolute FID is not comparable across sizes (different training iterations).

| Model | Method | FID ↓ | sFID ↓ | Prec. ↑ | Rec. ↑ |
|---|---|---|---|---|---|
| DiT-B/2 | Baseline | 11.52 | 4.87 | 0.719 | **0.554** |
| DiT-B/2 | RSTR† | **9.22** | **4.76** | **0.750** | 0.530 |
| DiT-S/2 | Baseline | 24.09 | 5.83 | 0.610 | **0.568** |
| DiT-S/2 | RSTR† | **20.00** | **5.56** | **0.646** | 0.553 |

*Table 9.* Effect of reference trajectory length $T_{\text{ref}}$ (DiT-XL/2, 512×512). †: without adaptive caching.

| Method | Steps | MACs↓ | IS↑ | FID↓ | sFID↓ | Prec.↑ | Rec.↑ |
|---|---|---|---|---|---|---|---|
| DDIM | 1000 | 1049.1 | 210.6 | 2.99 | 4.38 | 83.17 | 55.6 |
| DDIM | 50 | 52.45 | 203.8 | 3.20 | 4.53 | 83.27 | 56.4 |
| RSTR† ($T_{\text{ref}}=50$) | 50 | 33.56 | **229.6** | 2.84 | 4.40 | **83.80** | 56.0 |
| RSTR† ($T_{\text{ref}}=500$) | 50 | 34.09 | 225.0 | 2.77 | 4.39 | 83.49 | 56.2 |
| RSTR† ($T_{\text{ref}}=1000$) | 50 | **30.94** | 228.3 | **2.71** | **4.38** | 83.43 | **57.0** |

**Effect of Reference Trajectory Length**  Table 9 analyzes how reference trajectory length affects the quality of discovered schedules. Short references ($T_{\text{ref}}$=50) yield denser schedules (33.56T MACs) with FID 2.84, while longer references provide smoother optimization targets that help identify critical timesteps more accurately. At $T_{\text{ref}}$=1000, RSTR discovers the sparsest schedule (30.94T MACs) with the best FID (2.71), confirming that high-quality reference trajectories are crucial for effective evolutionary search.

**Adaptive vs. Uniform Rank Allocation**  Table 10 validates our adaptive rank allocation strategy. Uniform $r$=1024 achieves strong FID (2.92) but requires 55% more computation (34.68T vs 22.37T MACs). Lower uniform ranks reduce cost but degrade quality (FID 4.46 for $r$=512, 3.68 for $r$=256). RSTR discovers region-specific distributions achieving FID 3.01 with only 22.37T MACs and the highest precision (83.82%), demonstrating that different transformer regions require different calibration capacities under variable guidance. Additional results on PixArt-$\alpha$ in Appendix E.5 confirm generalizability across architectures.

*Table 10.* Adaptive vs. uniform rank allocation (DiT-XL/2, 512×512). †: without adaptive caching.

| Method | T | MACs↓ | IS↑ | FID↓ | sFID↓ | Pr.↑ | Re.↑ |
|---|---|---|---|---|---|---|---|
| DDIM | 50 | 52.45 | 203.8 | 3.25 | 4.53 | 83.27 | 56.4 |
| ICC | 50 | 33.43 | 200.0 | 3.73 | 5.39 | 83.30 | 55.6 |
| RSTR$^{\dagger}$ | 50 | 30.94 | **228.3** | **2.71** | **4.38** | 83.43 | **57.0** |
| + uni. $r$=1024 | 50 | 34.68 | 229.8 | 2.92 | 4.96 | 82.12 | 54.6 |
| + uni. $r$=512 | 50 | 26.16 | 205.0 | 4.46 | 5.19 | 78.19 | 55.4 |
| + uni. $r$=256 | 50 | 23.94 | 213.7 | 3.68 | 6.47 | 82.61 | 54.8 |
| **RSTR (adaptive)** | 50 | **22.37** | 228.1 | 3.01 | 4.63 | **83.82** | 54.9 |

*Table 11.* Per-step guidance leverage on DiT-XL/2 (ImageNet $256 \times 256$). For each step we apply guidance ($w$=5.0) at that step alone and report the intrinsic curvature $\kappa$ of the unguided trajectory and the curvature ratio $\kappa(\text{guided})/\kappa(\text{unguided})$.

| Step | Intrinsic $\kappa$ | Curvature ratio |
|---|---|---|
| 27 | 0.00004 | 7.3 |
| 24 | 0.00008 | 6.9 |
| 38 | 0.00003 | 6.0 |
| 39 | 0.00003 | 5.4 |
| 47 | 0.006 | 1.0 |

**Optimization Efficiency** Stage 1 optimization can be performed at reduced resolution since guidance schedules capture

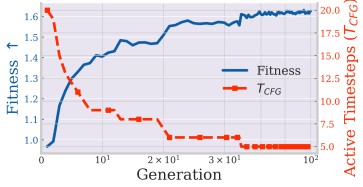

*Figure 5.* Stage 1 convergence behavior across generations.

semantic-level decisions rather than pixel-level details. For FLUX (native $1024{\times}1024$), we search at $128{\times}128$, completing in 1.5 hours on 4 H100 GPUs. As shown in Figure 5, fitness (balancing quality and sparsity) stabilizes within 20 generations while active CFG steps decrease. The discovered schedules transfer directly to full resolution without re-optimization.

### 5.4. Geometric Properties of the Discovered Positions

In this part, we discuss the geometric perspective and insights of what RSTR has discovered via its optimization. RSTR obtains its guidance schedule through evolutionary search, yet the discovered positions are not arbitrary: they reflect intrinsic geometric properties of the diffusion trajectory. The remove ablation in Table 6 shows that the high-scale steps (24, 27, 38, 39) are individually important for quality; here we ask what geometric property makes them effective, analyzing them on DiT-XL/2 through the geometry of the probability-flow ODE (PF-ODE) trajectory.

**Intrinsic trajectory curvature.** We analyze the PF-ODE trajectory under conditional-only (no-guidance) sampling and measure its intrinsic curvature $\kappa(t)$. The curvature varies by over $100\times$ across timesteps: it decreases monotonically from step 0 ($\kappa$=0.029) to a broad low-curvature region spanning roughly steps 20 to 42 ($\kappa < 0.0003$), then rises again toward step 48. The four positions 24, 27, 38, and 39 all fall within this low-curvature region.

**Guidance leverage.** To measure how strongly guidance affects the trajectory at each step in isolation, we apply $w$=5.0 at a single step only (all other steps conditional-only) and compute the curvature ratio $\kappa(\text{guided})/\kappa(\text{unguided})$, reported per step in Table 11. At the four positions 24, 27, 38, and 39, applying guidance bends the trajectory substantially (curvature ratio 5.4 to 7.3). In contrast, step 47, which sits at a point of high intrinsic curvature ($\kappa$=0.006), has a curva-

ture ratio of 1.0: guidance there does not alter the trajectory direction, because the trajectory's own bending dynamics already dominate. This indicates that RSTR follows the principle that *guidance is most effective where the trajectory is intrinsically straight*, and that the search successfully locates such steps.

**Guidance direction diversity.** Analyzing the guidance direction $\mathbf{g}(t) = \epsilon_c - \epsilon_u$, the four positions form two groups whose directions are mutually distinct (inter-group cosine similarity at most 0.30): $\{24, 27\}$ and $\{38, 39\}$. Each group steers the trajectory in a fundamentally different direction, so removing an entire group eliminates an irreplaceable guidance component, consistent with the quality drops in Table 6. Together, these analyses show that RSTR places guidance where the trajectory is intrinsically straight (low curvature, hence high leverage) and where the guidance directions split into non-redundant groups.

## 6. Conclusion

This paper addresses spatiotemporal redundancy in diffusion transformers. For temporal redundancy, we jointly optimize *when* to apply CFG (discrete skip patterns) and *what scale* to use (continuous values) via evolutionary search, discovering that sparse high-scale guidance can replace dense low-scale CFG. For spatial redundancy, we introduce adaptive rank allocation that assigns calibration capacities to transformer regions based on their sensitivity, enabling effective caching under variable guidance. Experiments on DiT-XL/2, PixArt-$\alpha$, FLUX, and Qwen-Image confirm 50% to 70% computational savings while maintaining or improving generation quality. The discovered schedules also generalize across CFG strengths.

## Acknowledgements

This work was supported in part by the National Science Foundation (NSF) under awards SES-2521631, CCF-2428108, and OAC-2403090. Any errors and opinions are not those of the NSF and are attributable solely to the author(s).

## Impact Statement

This work advances the efficiency of diffusion transformers through optimized guidance scheduling and adaptive caching. While our contributions are primarily technical, improving inference speed without sacrificing generation quality, we acknowledge that advances in generative models can have broader societal implications. These may include beneficial applications such as democratizing access to high-quality image synthesis and reducing energy consumption, as well as potential concerns regarding synthetic media. Our method does not introduce new generation capabilities but accelerates existing models. We encourage ongoing discussion about the responsible deployment of such technologies.

We also note several technical limitations of our approach. The discovered schedules are specific to a given model architecture and sampling configuration, and may require re-optimization when these change. The two stages are optimized sequentially rather than jointly, which may leave room for further improvement. We leave a broader characterization of robustness across prompt domains and model families to future work.

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

## APPENDIX

# Appendix Table of Contents

# A. Guidance Schedule Search Space Analysis

We provide theoretical justification for RSTR's joint optimization of guidance schedules. All notation follows the main text.

## A.1. Problem Formulation

Consider the DDIM sampling process with CFG. At timestep $t$, the guided noise prediction is

$$\tilde{\epsilon}_\theta(x_t, c, t) = \epsilon_\theta(x_t, \varnothing, t) + w_t \cdot [\epsilon_\theta(x_t, c, t) - \epsilon_\theta(x_t, \varnothing, t)], \tag{9}$$

where $w_t \geq 0$ is the guidance scale. A **guidance schedule** is a vector $\mathbf{w} = [w_T, \ldots, w_1]^\top \in \mathbb{R}^T$.

**Definition A.1** (Guidance Schedule Spaces). We define the following schedule spaces:

$$\Omega_{\text{const}} = \{w \cdot \mathbf{1}_T : w > 1\}, \tag{10}$$

$$\Omega_{\text{interval}} = \{\mathbf{w} : w_t \in \{1, w\}, \ w_t = w \Leftrightarrow t \in [t_{\text{lo}}, t_{\text{hi}}]\}, \tag{11}$$

$$\Omega_{\text{scheduler}} = \{\mathbf{w} : w_t = f_\phi(t), \ w_t > 1 \ \forall t\}, \tag{12}$$

$$\Omega_{\text{RSTR}} = [1, w_{\max}]^T. \tag{13}$$

Here $w_t = 1$ corresponds to conditional-only prediction (no guidance), while $w_t > 1$ applies CFG. The space $\Omega_{\text{const}}$ corresponds to standard CFG (Ho & Salimans, 2022), $\Omega_{\text{interval}}$ to Limited Interval (Kynkäänniemi et al., 2024) where guidance is active only within $[t_{\text{lo}}, t_{\text{hi}}]$, and $\Omega_{\text{scheduler}}$ to parametric schedulers (Wang et al., 2024) that apply guidance at all timesteps.

**Proposition A.2** (Search Space Inclusion). *The following strict inclusions hold:*

$$\Omega_{\text{const}} \subsetneq \Omega_{\text{interval}} \subsetneq \Omega_{\text{RSTR}}, \quad \Omega_{\text{const}} \subsetneq \Omega_{\text{scheduler}} \subsetneq \Omega_{\text{RSTR}}. \tag{14}$$

*Proof.* The inclusion $\Omega_{\text{const}} \subsetneq \Omega_{\text{interval}}$ follows from setting $t_{\text{lo}} = 1, t_{\text{hi}} = T$. For $\Omega_{\text{interval}} \subsetneq \Omega_{\text{RSTR}}$: any interval schedule with $w \leq w_{\max}$ lies in $[1, w_{\max}]^T$, but $\Omega_{\text{RSTR}}$ also contains non-contiguous patterns (e.g., $\mathbf{w}$ with $w_t > 1$ only for $t \in \{5, 15, 25\}$) and variable scales within active timesteps. Similarly, $\Omega_{\text{scheduler}} \subsetneq \Omega_{\text{RSTR}}$ since schedulers require $w_t > 1 \ \forall t$, while $\Omega_{\text{RSTR}}$ permits $w_t = 1$. $\square$

**Corollary A.3.** *For any loss function $\mathcal{L} : \mathbb{R}^T \to \mathbb{R}$,*

$$\min_{\mathbf{w} \in \Omega_{\text{RSTR}}} \mathcal{L}(\mathbf{w}) \leq \min \left\{ \min_{\mathbf{w} \in \Omega_{\text{interval}}} \mathcal{L}(\mathbf{w}), \ \min_{\mathbf{w} \in \Omega_{\text{scheduler}}} \mathcal{L}(\mathbf{w}) \right\}. \tag{15}$$

## A.2. Motivation for Sparse Variable Schedules

The design of RSTR's search space is motivated by two complementary findings from prior work:

- **Sparsity is sufficient.** Kynkäänniemi et al. (2024) demonstrate that CFG at high noise levels causes mode collapse toward "template images," while CFG at low noise levels has minimal effect on outputs. This establishes that guidance is beneficial only within a limited interval, and a proper subset of timesteps $\mathcal{S}^* \subsetneq \{1, \ldots, T\}$ suffices.
- **Variable scales are beneficial.** Wang et al. (2024) show that time-varying scales (e.g., monotonically increasing) outperform constant guidance, mitigating excessive guidance at early steps.

These methods optimize different aspects: $\Omega_{\text{interval}}$ optimizes *when* to apply guidance but uses constant scales, while $\Omega_{\text{scheduler}}$ optimizes *what* scale to use but keeps all timesteps active. Neither jointly optimizes both. RSTR's evolutionary search discovers schedules in $\Omega_{\text{RSTR}}$ that are simultaneously sparse (only a subset $\mathcal{S} \subset \{1, \ldots, T\}$ has $w_t > 1$) and variable (different $t \in \mathcal{S}$ may have different $w_t$).

*Remark* A.4. The discovered active timesteps vary across optimization runs due to multiple local optima in the fitness landscape. The consistent empirical finding across DiT-XL/2, PixArt-$\alpha$, FLUX, and Qwen-Image is that schedules with $|\mathcal{S}|/T \in [0.15, 0.50]$ suffice to match or exceed full CFG quality.

## B. Denoising Deviations Under Variable Guidance

### B.1. Mathematical Analysis

The sparse guidance schedules from Stage 1 introduce denoising deviations that affect caching effectiveness. We analyze two primary scenarios that arise from our variable guidance patterns.

**Scenario 1: Guidance Scale Variation.** When consecutive timesteps both apply CFG but with different scales ($w_t^* > \tau$ and $w_{t-1}^* > \tau$), we derive the resulting deviation.

Starting from the DDIM update (Song et al., 2020a) with deterministic sampling ($\sigma_t = 0$):

$$x_{t-1} = \sqrt{\frac{\bar{\alpha}_{t-1}}{\bar{\alpha}_t}} x_t + \beta_{t-1,t} \cdot \tilde{\epsilon}_\theta(x_t, t) \tag{16}$$

where $\beta_{t-1,t} = \sqrt{1 - \bar{\alpha}_{t-1}} - \sqrt{\frac{\bar{\alpha}_{t-1}(1-\bar{\alpha}_t)}{\bar{\alpha}_t}}$.

With CFG, the noise prediction is:

$$\tilde{\epsilon}_\theta(x_t, t) = \epsilon_u(x_t, t) + w_t^*[\epsilon_c(x_t, t) - \epsilon_u(x_t, t)] \tag{17}$$

When guidance scale changes from $w_t^*$ to $w_{t-1}^*$, the deviation becomes:

$$\Delta x_{t-1}^{\text{scale}} = \beta_{t-1,t}(w_{t-1}^* - w_t^*)[\epsilon_c(x_t, t) - \epsilon_u(x_t, t)] \tag{18}$$

For the approximation in Equation 7 of the main text, we use $\beta_{t-1,t} \approx \sqrt{\bar{\alpha}_{t-1}/\bar{\alpha}_t}$ for clarity.

**Scenario 2: Guidance Mode Switching.** When timestep $t$ uses CFG ($w_t^* > \tau$) but timestep $t-1$ switches to conditional-only ($w_{t-1}^* < \tau$), this is equivalent to switching from $w_t^*$ to $w_{t-1}^* = 1$ (since conditional-only means $\tilde{\epsilon} = \epsilon_c$).

Following the same framework, the switching deviation is:

$$\Delta x_{t-1}^{\text{switch}} = \beta_{t-1,t}(1 - w_t^*)[\epsilon_c(x_t, t) - \epsilon_u(x_t, t)] \tag{19}$$

Note that when $w_t^* > 1$, this deviation has opposite sign compared to Scenario 1, creating an abrupt trajectory change.

**Impact on Feature Caching.** These deviations directly affect cached feature validity. Equations 18 and 19 reveal that variable guidance creates trajectory discontinuities that exceed uniform calibration's correction capacity, motivating our adaptive rank allocation in Stage 2.

### B.2. Empirical Validation

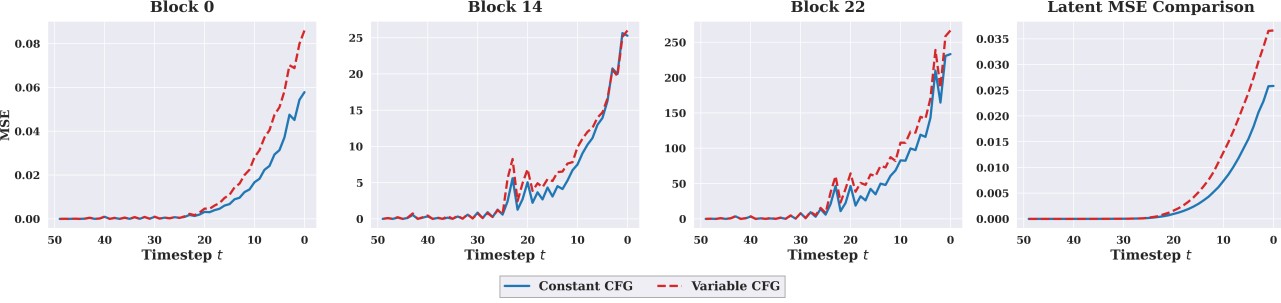

*Figure 6.* **Variable guidance patterns introduce feature discrepancies that degrade standard caching.** The first three subplots compare feature reconstruction errors (MSE) across representative transformer blocks in DiT-XL/2. While constant CFG (blue) maintains moderate error levels, the sparse guidance schedule (red) introduces significantly larger discrepancies. The rightmost panel highlights the cumulative impact, showing that these block-wise errors propagate to increase the MSE of the final predicted latent $x_0$.

To empirically validate the theoretical deviations derived above, we conducted experiments measuring feature reconstruction errors under different guidance patterns. Figure 6 compares feature consistency between constant CFG ($w = 1.5$) and

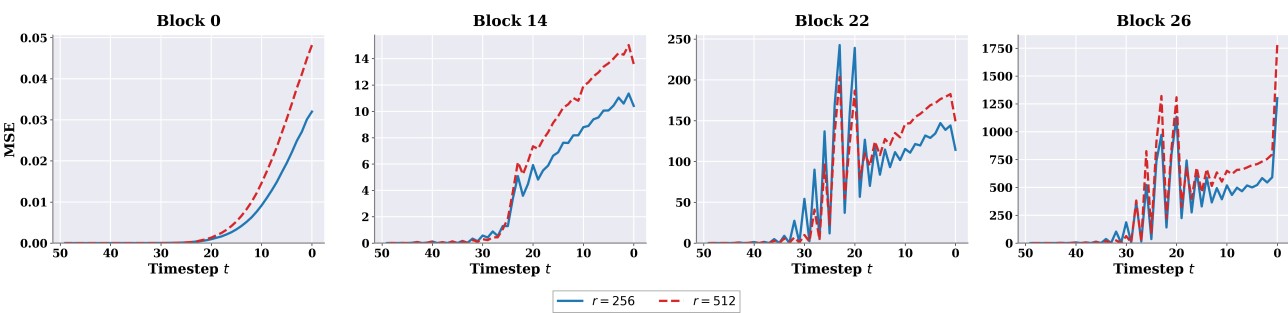

*Figure 7.* **Uniform rank allocation fails to address heterogeneous block sensitivity.** Comparison of feature reconstruction errors between uniform rank $r = 256$ (blue) and $r = 512$ (red). The results reveal conflicting requirements across the network: while the higher rank $r = 512$ is necessary to suppress the severe error spikes in Block 22, it introduces higher errors in Blocks 0, 14, and 26 compared to $r = 256$. This demonstrates that no single uniform rank is optimal for all blocks.

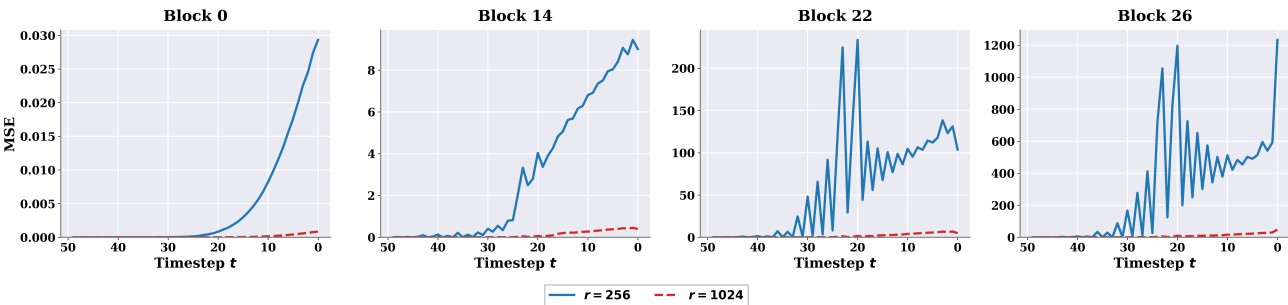

*Figure 8.* **High-capacity calibration consistently minimizes feature reconstruction errors.** This figure compares mean squared error between uniform rank $r = 256$ (blue) and $r = 1024$ (red). As expected, the high-capacity rank $r = 1024$ universally yields lower errors across all transformer blocks. This validates that sufficient calibration capacity can effectively correct feature deviations, serving as a performance upper bound with high computational cost.

our sparse guidance schedule from Stage 1. The results reveal that variable guidance introduces substantially higher reconstruction errors across representative transformer blocks. While Block 0 shows relatively small differences, the gap widens progressively in deeper blocks, where Block 14 and Block 22 exhibit significantly larger discrepancies under variable guidance. The rightmost panel shows the cumulative impact on the final predicted latent $x_0$: variable guidance causes approximately 40% higher MSE compared to the constant baseline, confirming that block-wise reconstruction errors propagate and accumulate through the network to degrade generation quality.

While incremental calibration can mitigate these errors, a critical question remains: should all transformer blocks use the same calibration rank? Figures 7 and 8 address this by comparing feature reconstruction errors under uniform ranks $r \in \{256, 512, 1024\}$. Although the high-capacity $r = 1024$ consistently yields the lowest error as expected, the comparison between $r = 256$ and $r = 512$ reveals a counter-intuitive insight. Specifically, Block 0 and Block 26 exhibit lower MSE with $r = 256$ compared to $r = 512$, demonstrating that certain blocks do not require excessive incremental information for effective correction. This heterogeneous pattern confirms that uniform calibration is inefficient, which directly motivates our region-aware adaptive rank allocation strategy in Stage 2.

## C. Model-Specific Configurations

We provide the specific hyperparameter configurations for each model below. A summary is provided in Table 12.

*Table 12.* Summary of hyperparameters used in RSTR optimization across different models.

| Hyperparameter | DiT-XL/2 | PixArt-$\alpha$ | FLUX | Qwen-Image |
|---|---|---|---|---|
| Sampler | DDIM | DPM-Solver | Euler | Euler A |
| Guidance Mode | Standard CFG | Standard CFG | True CFG | Standard CFG |
| Steps ($T$) | 30/50 | 20 | 20 | 20 |
| CFG Scale | 1.5 | 4.5 | 1.5 | 4.0 |
| *Stage 1 (Evolution)* | | | | |
| Search Resolution | $256^2$ / $512^2$ | $256^2$ | $128^2$ | $256^2$ |
| Population ($P$) | 16 | 32 | 32 | 32 |
| Generations ($G$) | 10 | 15 | 15 | 15 |
| Reference Steps ($T_{\text{ref}}$) | 1000 | 1000 | 100 | 100 |
| Calibration Batch Size | 32 | 40 | 8 | 32 |
| Max CFG ($w_{\max}$) | 10.0 | 10.0 | 10.0 | 15.0 |
| CFG Threshold ($\tau$) | 1.5 | 1.0 | 1.5 | 4.0 |
| *Stage 2 (Caching)* | | | | |
| Regions ($K$) | 7 | 7 | 4 | 10 |
| Calibration Images | 50,000 | 30,000 | 5,000 | 10,000 |
| Rank Range | $[16, 512]$ | $[16, 256]$ | $[16, 512]$ | $[16, 1024]$ |

**DiT-XL/2.** We evaluate the class-conditional DiT-XL/2 on ImageNet. Following standard protocols, we use the DDIM sampler with 30–50 denoising steps and a classifier-free guidance scale of $w = 1.5$. In Stage 1, to encourage broad exploration of guidance intensities, we set the search bound $w_{\max} = 10.0$ with a sparsity threshold $\tau = 1.5$. The optimization runs with $P = 16$ and $G = 10$ using a calibration batch of 32 diverse class prompts. In Stage 2, to capture fine-grained feature deviations, we increase the partition granularity to $K = 7$ regions. We utilize a large calibration set of 50,000 images to ensure robust rank allocation, searching within $[16, 512]$.

**PixArt-$\alpha$.** For text-to-image generation, we use the PixArt-$\alpha$ model with the DPM-Solver (20 steps) and a guidance scale of $w = 4.5$. Stage 1 optimization is performed at $256 \times 256$ resolution with $w_{\max} = 10.0$ and $\tau = 1.0$. We use a population of $P = 32$ for $G = 15$ generations to handle the text-image search space. In Stage 2, we similarly partition the model into $K = 7$ regions and use 30,000 calibration images to determine optimal ranks within $[16, 256]$.

**FLUX.** We employ the FLUX.1-dev model. For sampling, we utilize the Euler sampler (specifically `FlowMatchEulerDiscreteScheduler`). For guidance, we enable True Classifier-Free Guidance with a scale of $w = 1.5$. Stage 1 search is conducted at $128 \times 128$ with $w_{\max} = 10.0$ and $\tau = 1.5$, utilizing a calibration batch of 8 to accelerate evaluation. In Stage 2, we partition the massive 12B model into $K = 4$ regions and search for ranks in $[16, 512]$ using 5,000 images.

**Qwen-Image.** We use the Euler Ancestral sampler with 20 steps and a guidance scale of $w = 4.0$. Stage 1 settings use a wider search range with $w_{\max} = 15.0$ and a higher threshold $\tau = 4.0$ to accommodate the model's sensitivity. In Stage 2, given the depth and complexity of LMMs, we use a fine-grained partition of $K = 10$ regions. We expand the rank search range to $[16, 1024]$ and use 10,000 calibration images to ensure alignment.

## D. Optimization Cost Analysis

A key advantage of RSTR is that the optimization is a **one-time offline process**. Once the optimal schedule $\mathbf{w}^*$ and rank configuration $\mathbf{r}^*$ are discovered for a specific model version, they can be deployed indefinitely without further overhead.

We benchmark the optimization timing for all four evaluated models on their respective hardware; the detailed per-model timing breakdown is provided in our code repository at https://github.com/rusu4943/RSTR.

**Guidance-driven caching protocol.** We adapt the caching strategy to handle variable guidance patterns from Stage 1. The cache schedule dynamically combines intervals of $N \in \{1, 2, 3\}$ depending on the guidance pattern: when consecutive timesteps require CFG transitions (from $w_t = 1$ to $w_{t-1} > 1$), we must perform full computation since the unconditional

branch was never executed. Overall, the adaptive schedule achieves computation savings equivalent to a uniform $N = 2$ interval while ensuring correct CFG application.

## E. Additional Experimental Results

### E.1. Analysis of Guidance Scale Threshold ($\tau$)

The CFG threshold $\tau$ determines when guidance is applied: timesteps with $w_t \geq \tau$ perform full CFG, while those below use only conditional prediction. Together with the reference CFG scale $w_{\text{const}}$ used in Stage 1, this threshold controls the sparsity of the discovered guidance schedule.

Figure 9 illustrates this relationship on Qwen-Image, where the Stage 1 reference trajectory uses $w_{\text{const}} = 4.0$. At low thresholds ($\tau = 2$), approximately 14 timesteps remain active. As $\tau$ increases to 4, this gradually decreases to around 10 active steps. Notably, a sharp transition occurs between $\tau = 4$ and $\tau = 5$, where active steps drop dramatically from 10 to fewer than 2. Beyond $\tau = 5$, almost no timesteps remain active. This sharp transition suggests that evolutionary optimization concentrates most guidance values slightly above the reference scale $w_{\text{const}} = 4.0$.

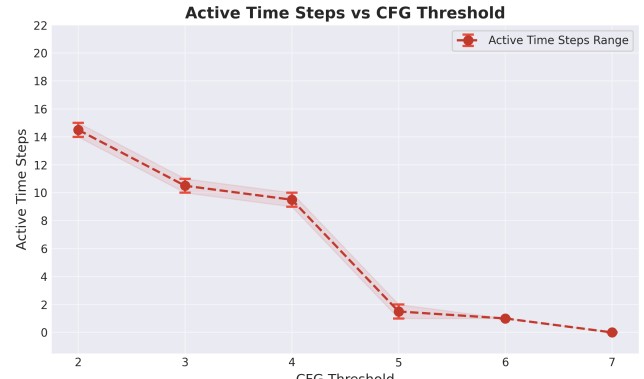

*Figure 9.* Relationship between CFG threshold $\tau$ and the number of active guidance steps on Qwen-Image. A sharp transition occurs between $\tau = 4$ and $\tau = 5$, indicating that most optimized guidance values concentrate in this range.

### E.2. Analysis of Maximum Guidance Scale ($w_{\text{max}}$)

The maximum guidance scale ($w_{\text{max}}$) defines the boundary of our evolutionary search space. We conducted a sensitivity analysis on the **Qwen-Image** model by varying $w_{\text{max}}$ from 5.0 to 17.5. To rigorously evaluate robustness, we compared RSTR against the standard Euler Ancestral (Euler A) baselines of Qwen-Image. Specifically, we examine performance at the **original** scale (1.0×) as well as scaled versions (0.75× and 1.5×). Note that for the baseline, the **original** scale corresponds to a constant CFG = 4.0.

Figure 10 presents the ImageReward landscape. Our analysis reveals three key insights:

**Optimal Stability at $w_{\text{max}} = 15$.** Contrary to lower settings where performance fluctuates significantly with scaling, the configuration $w_{\text{max}} = 15$ exhibits remarkable stability. As shown in Figure 10, this setting achieves the highest "performance floor". Even when guidance is reduced to 0.75× (orange triangle), the ImageReward remains approximately 1.19, which is comparable to the **original** performance of other settings. This indicates that $w_{\text{max}} = 15$ yields the most robust schedule, capable of maintaining high generation quality across a wide range of adjustments during inference.

**Surpassing the Computationally Demanding Baseline.** A comparison with the Euler A baseline at 50 steps (the rightmost gray line, ImageReward $\approx 1.21$) highlights the efficiency of our method. The optimized schedule of RSTR at $w_{\text{max}} = 15$, particularly when scaled by 1.5× (green triangle), achieves an ImageReward of 1.22. This

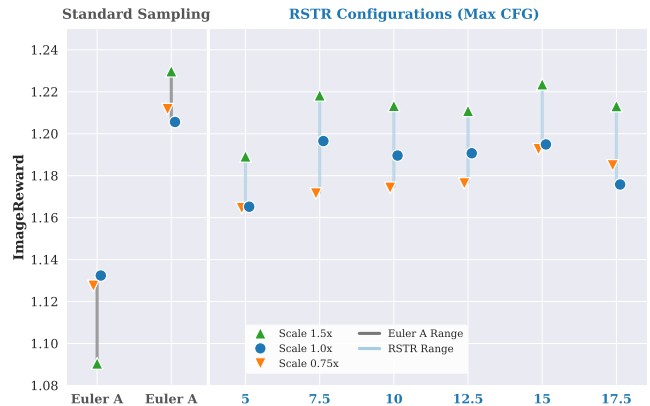

*Figure 10.* **Sensitivity Analysis on Qwen-Image.** Comparison of RSTR against Euler A baselines (gray vertical lines). While the baseline degrades at high CFG (6.0), RSTR demonstrates superior stability. Specifically, the configuration $w_{\text{max}} = 15$ achieves the best overall performance, maintaining high quality even when guidance is reduced (orange triangles).

effectively matches or exceeds the quality of the baseline with 50 steps while requiring only 15 steps. This confirms that sparse and high magnitude guidance can substitute for dense sampling steps.

**Consistent Superiority over the Standard Baseline.** We observe that the results of RSTR **at the original scale** (blue circles) consistently outperform the corresponding Euler A baseline at 15 steps (the leftmost blue circle at $\approx 1.13$) across all $w_{\max}$ settings. Unlike the baseline, which suffers from degradation caused by artifacts when forced to high CFG (6.0), the learned schedules of RSTR benefit from scaling up. This suggests that our evolutionary optimization successfully filters out harmful guidance signals while retaining the beneficial ones.

### E.3. Guidance Schedule Transferability

A robust guidance schedule must generalize beyond its calibration point. We evaluate transferability by applying multiplicative scaling factors $k$ to the optimized schedule $\mathbf{w}^*$ (i.e., $k \cdot \mathbf{w}^*$) and comparing against constant CFG baselines scaled by the same factor.

*Table 13.* Guidance transferability on DiT-XL/2 (ImageNet $256 \times 256$, $w_{\text{ref}}$=1.5). Equivalent constant CFG scales are 1.5, 5.0, and 7.5 for $1.0\times$, $3.3\times$, and $5.0\times$ respectively.

| Method | Steps | CFG-Steps | FID ↓ | | | sFID ↓ | | |
|---|---|---|---|---|---|---|---|---|
| | | | $1.0\times$ | $3.3\times$ | $5.0\times$ | $1.0\times$ | $3.3\times$ | $5.0\times$ |
| DDIM | 50 | 50 | 2.23 | 16.39 | 19.71 | 4.29 | 15.53 | 19.95 |
| RSTR† | 50 | 8 | **2.10** | **9.20** | **11.76** | **4.29** | **9.23** | **13.20** |

*Table 14.* Guidance transferability on FLUX (GenEval $512 \times 512$, $w_{\text{ref}}$=1.5). Equivalent constant CFG scales are 1.5, 3.0, and 4.5 for $1\times$, $2\times$, and $3\times$ respectively.

| Method | Scale | Steps | CFG-Steps↓ | Pos. | Colors | Count | Col-Attr | Two-Obj | Single-Obj | Overall |
|---|---|---|---|---|---|---|---|---|---|---|
| FLUX | $\times 1$ | 50 | 50 | 19.50 | 80.32 | 77.19 | 49.75 | 86.11 | 98.75 | 68.60 |
| FLUX | $\times 2$ | 50 | 50 | 23.75 | 73.67 | 75.31 | 40.25 | 82.32 | 95.94 | 65.20 |
| FLUX | $\times 3$ | 50 | 50 | 20.75 | 53.72 | 62.81 | 21.25 | 72.22 | 90.31 | 53.51 |
| RSTR† | $\times 1$ | 20 | 8 | 21.25 | 80.32 | 68.44 | 51.50 | 84.85 | 98.44 | 67.46 |
| RSTR† | $\times 2$ | 20 | 8 | 20.50 | 79.52 | 65.31 | 36.75 | 79.04 | 98.75 | 63.31 |
| RSTR† | $\times 3$ | 20 | 8 | 16.50 | 72.34 | 56.88 | 23.25 | 61.87 | 96.25 | 54.51 |

*Table 15.* Guidance transferability on Qwen-Image (drawbench $1024 \times 1024$, $w_{\text{ref}}$=4.0). Equivalent constant CFG scales are 3.0, 4.0, and 6.0 for $0.75\times$, $1.0\times$, and $1.5\times$ respectively.

| Method | Steps | CFG-Steps | ImageReward ↑ | | | CLIP Score ↑ | | |
|---|---|---|---|---|---|---|---|---|
| | | | $0.75\times$ | $1.0\times$ | $1.5\times$ | $0.75\times$ | $1.0\times$ | $1.5\times$ |
| Qwen-Image | 50 | 50 | 1.212 | 1.206 | 1.230 | 29.01 | 29.13 | 29.15 |
| Qwen-Image | 15 | 15 | 1.128 | 1.132 | 1.091 | 28.94 | **28.90** | 28.62 |
| RSTR† | 20 | 10 | **1.193** | **1.195** | **1.224** | **28.95** | 28.89 | **28.91** |

**DiT-XL/2.** Table 13 reveals a dramatic contrast in stability. When scaling by $3.3\times$, constant CFG collapses (FID 2.23 $\rightarrow$ 16.39) while RSTR† degrades gracefully (2.10 $\rightarrow$ 9.20). At $5\times$ scaling, the gap widens further: DDIM reaches FID 19.71 while RSTR† maintains 11.76.

**FLUX.** Table 14 shows similar robustness on GenEval. Under aggressive $3\times$ scaling, RSTR (54.51) slightly outperforms constant CFG (53.51) while using only 8 guidance steps versus 50. The Colors metric particularly benefits from RSTR's sparse schedule, maintaining 72.34 compared to 53.72 for the baseline.

**Qwen-Image.** Table 15 demonstrates that RSTR exhibits positive transferability. While the baseline degrades at high guidance (ImageReward drops from 1.132 to 1.091 at $1.5\times$), RSTR improves (1.195 $\rightarrow$ 1.224), approaching the compute-intensive 50-step baseline. This confirms that our sparse schedules effectively filter harmful guidance signals while retaining beneficial ones.

### E.4. Cross-Sampler Transfer

The Stage-1 schedules are discovered using a particular sampler (DPM-Solver for PixArt-$\alpha$). To test whether a schedule is tied to its search-time sampler, we apply the *same* optimized schedule, without re-optimization, to SA-Solver (Xue et al.,

2023), a stochastic (SDE) predictor-corrector sampler from a family different from the ODE-based DPM-Solver used during the search.

*Table 16.* Cross-sampler transfer on PixArt-$\alpha$ (COCO 30K, 256 $\times$ 256, 20 steps). The RSTR$^{\dagger}$ schedule is optimized with DPM-Solver and applied to SA-Solver without re-optimization. RSTR$^{\dagger}$ denotes the optimized guidance schedule without adaptive caching. The DPM-Solver rows are reproduced from Table 2.

| Method | FID ↓ |
|---|---|
| Baseline DPM-Solver (CFG=4.5) | 24.60 |
| RSTR$^{\dagger}$ DPM-Solver | 22.69 |
| Baseline SA-Solver (CFG=4.5) | 23.87 |
| RSTR$^{\dagger}$ SA-Solver | **18.83** |

On SA-Solver, the transferred schedule improves FID from 23.87 to **18.83** ($-5.04$), an apples-to-apples comparison on the same sampler. The RSTR$^{\dagger}$+SA-Solver combination yields the best result among the four configurations in Table 16. This indicates that the optimized schedule captures per-timestep guidance importance that is not specific to the sampler used during the search, transferring cleanly from an ODE solver to a stochastic SDE solver.

### E.5. Adaptive vs. Uniform Rank Allocation

We validate our adaptive rank allocation strategy on both DiT-XL/2 and PixArt-$\alpha$ to demonstrate its generalizability across different architectures and datasets.

### DiT-XL/2 (ImageNet).

*Table 17.* Ablation study of rank allocation strategies on DiT-XL/2 (ImageNet 512×512). †denotes without adaptive caching.

| Method | Steps | MACs↓ | IS↑ | FID↓ | sFID↓ | Prec.↑ | Rec.↑ |
|---|---|---|---|---|---|---|---|
| DDIM | 50 | 52.45 | 203.8 | 3.25 | 4.53 | 83.27 | 56.4 |
| ICC | 50 | 33.43 | 200.0 | 3.73 | 5.39 | 83.30 | 55.6 |
| RSTR$^{\dagger}$ | 50 | 30.94 | 228.3 | **2.71** | **4.38** | 83.43 | **57.0** |
| RSTR w/ uni. $r$=1024 | 50 | 34.68 | **229.8** | 2.92 | 4.96 | 82.12 | 54.6 |
| RSTR w/ uni. $r$=512 | 50 | 26.16 | 205.0 | 4.46 | 5.19 | 78.19 | 55.4 |
| RSTR w/ uni. $r$=256 | 50 | 23.94 | 213.7 | 3.68 | 6.47 | 82.61 | 54.8 |
| **RSTR** | 50 | **22.37** | 228.1 | 3.01 | 4.63 | **83.82** | 54.9 |

Table 17 demonstrates that uniform $r$=1024 achieves the best FID among uniform configurations (2.92) but requires 55% more computation (34.68T vs 22.37T MACs) than RSTR. Lower uniform ranks ($r$=512, $r$=256) reduce cost but severely degrade quality (FID 4.46 and 3.68). RSTR discovers region-specific rank distributions that achieve FID 3.01 with only 22.37T MACs and the highest precision (83.82), demonstrating that different transformer regions require different calibration levels under variable guidance patterns.

### PixArt-$\alpha$ (MSCOCO).

*Table 18.* Ablation study of rank allocation strategies on PixArt-$\alpha$ (MSCOCO 256×256). †denotes without adaptive caching.

| Method | Steps | MACs↓ | FID↓ | CLIP↑ |
|---|---|---|---|---|
| DPM-Solver | 1000 | 336.11 | 22.97 | 16.42 |
| DPM-Solver | 20 | 6.72 | 24.60 | 16.31 |
| RSTR$^{\dagger}$ | 20 | 4.37 | 22.69 | 16.39 |
| RSTR w/ uni. $r$=32 | 20 | 2.63 | 20.05 | 16.49 |
| RSTR w/ uni. $r$=64 | 20 | 2.70 | 19.92 | **16.52** |
| **RSTR** | 20 | **2.67** | **19.27** | 16.48 |

Table 18 confirms similar trends on PixArt-$\alpha$. Uniform ranks ($r$=32, $r$=64) show varying trade-offs between cost and quality. RSTR achieves the best FID (19.27) with minimal MACs (2.67T), surpassing both the 20-step (FID 24.60) and

1000-step (FID 22.97) DPM-Solver baselines while using 60% less computation than the 20-step baseline. These results across different architectures validate the generalizability of our adaptive rank allocation strategy.

### E.6. Effect of Region Count on Adaptive Rank Allocation

In Stage 2, we partition the $N$ transformer blocks into $K$ regions for adaptive rank allocation. The choice of $K$ controls the granularity of calibration: $K = 1$ corresponds to uniform rank across all blocks, while larger $K$ allows finer-grained adaptation to block-specific requirements.

Figure 11 shows the effect of $K$ on DiT-XL/2 (28 blocks). FID improves significantly from $K = 1$ (2.12) to $K = 4$ (2.05), and further to $K = 7$ (2.03). However, increasing to $K = 14$ yields no additional benefit (FID 2.04), indicating that 7 regions sufficiently capture the heterogeneous calibration requirements across transformer blocks. Based on this analysis, we use $K = 7$ for DiT-XL/2 in our main experiments.

### E.7. Compatibility with INT8 Quantization

RSTR reduces computation through temporal optimization (sparse guidance schedules) and spatial optimization (adaptive rank allocation), which is orthogonal to

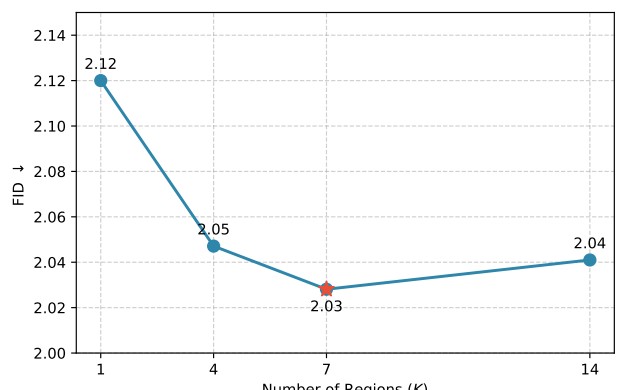

*Figure 11.* Impact of region count $K$ on adaptive rank allocation for DiT-XL/2 (ImageNet 256×256). $K = 7$ achieves the best FID (2.03), while further increasing to $K = 14$ yields no additional benefit.

quantization-based acceleration. We investigate whether these techniques can be combined for additional speedups.

*Table 19.* Quantization compatibility on DiT-XL/2 (ImageNet 256×256) with INT8. $T$: Steps, $N_{\mathbf{cfg}}$: CFG-Steps. $\dagger$: w/o cache.

| Prec. | Method | $T$ | $N_{\mathrm{cfg}} \downarrow$ | IS↑ | FID↓ | sFID↓ |
|---|---|---|---|---|---|---|
| FP16 | DDIM | 50 | 50 | 239.4 | 2.23 | 4.29 |
| | RSTR[†] | 50 | 8 | 263.0 | 2.10 | 4.29 |
| INT8 | DDIM | 50 | 50 | 199.9 | 4.61 | 9.17 |
| | RSTR[†] | 50 | 8 | **225.4** | **3.61** | **8.55** |

Table 19 shows that RSTR maintains its relative advantages under INT8 quantization. Both methods experience quality degradation from FP16 to INT8: DDIM's FID increases from 2.23 to 4.61, while RSTR[†]'s FID increases from 2.10 to 3.61. Crucially, RSTR[†] consistently outperforms DDIM under the same precision—in INT8, RSTR[†] achieves better FID (3.61 vs 4.61) and IS (225.4 vs 199.9) while using only 8 CFG steps. This confirms that RSTR is orthogonal to precision-reduction techniques and can be deployed alongside quantization for compounded acceleration benefits.

## F. Additional Qualitative Results

Figure 12 provides visual comparisons across 15 ImageNet classes on DiT-XL/2, demonstrating that RSTR achieves comparable visual quality to standard DDIM while using only 35% of the computational budget.

We further evaluate RSTR on text-to-image models with both short and long prompts. Figure 13 and Figure 14 show qualitative comparisons on FLUX and PixArt-$\alpha$ respectively, where RSTR maintains visual fidelity comparable to baselines across diverse prompt complexities.

To demonstrate the generalization of our optimized guidance schedules, Figure 17 show results when scaling the CFG strength beyond the training configuration. RSTR maintains stable generation quality across different CFG scales, indicating that the discovered sparse guidance patterns transfer well to varying guidance intensities.

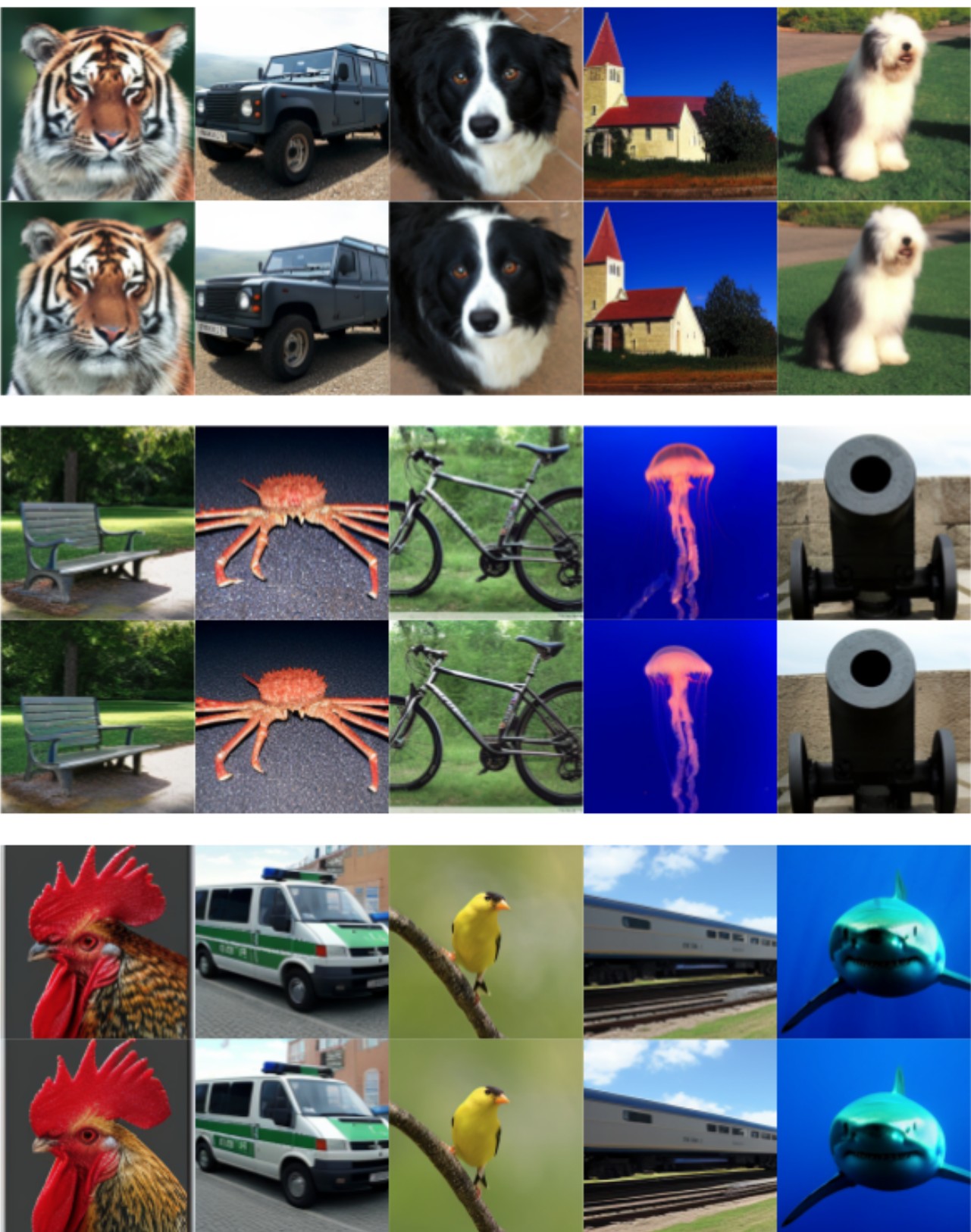

*Figure 12.* Visual quality comparison between standard DDIM (50 steps, 100% MACs) and RSTR (50 steps, 35% MACs) on DiT-XL/2 at 256×256 resolution across diverse ImageNet classes. Each pair shows DDIM (top) and RSTR (bottom) outputs from identical initial noise.

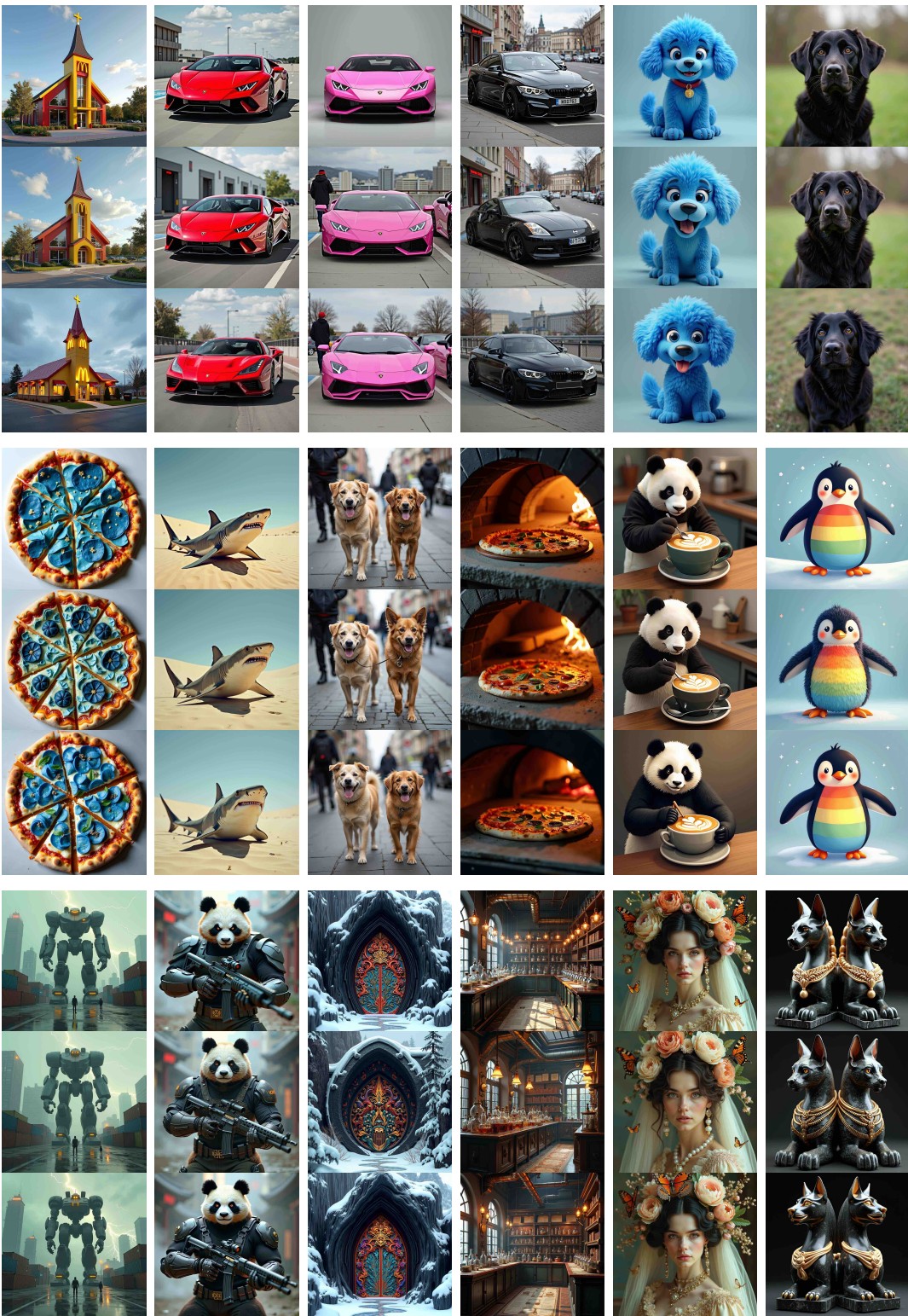

*Figure 13.* Visualization results of FLUX using short and long prompts. For each prompt, three methods are compared vertically: FLUX baseline (50 steps, True CFG=1.5) on top, TaylorSeer (N=3, O=2) in the middle, and RSTR (ours) at the bottom. The first two row-triplets are generated with short prompts, while the last row-triplet is produced using long prompts.

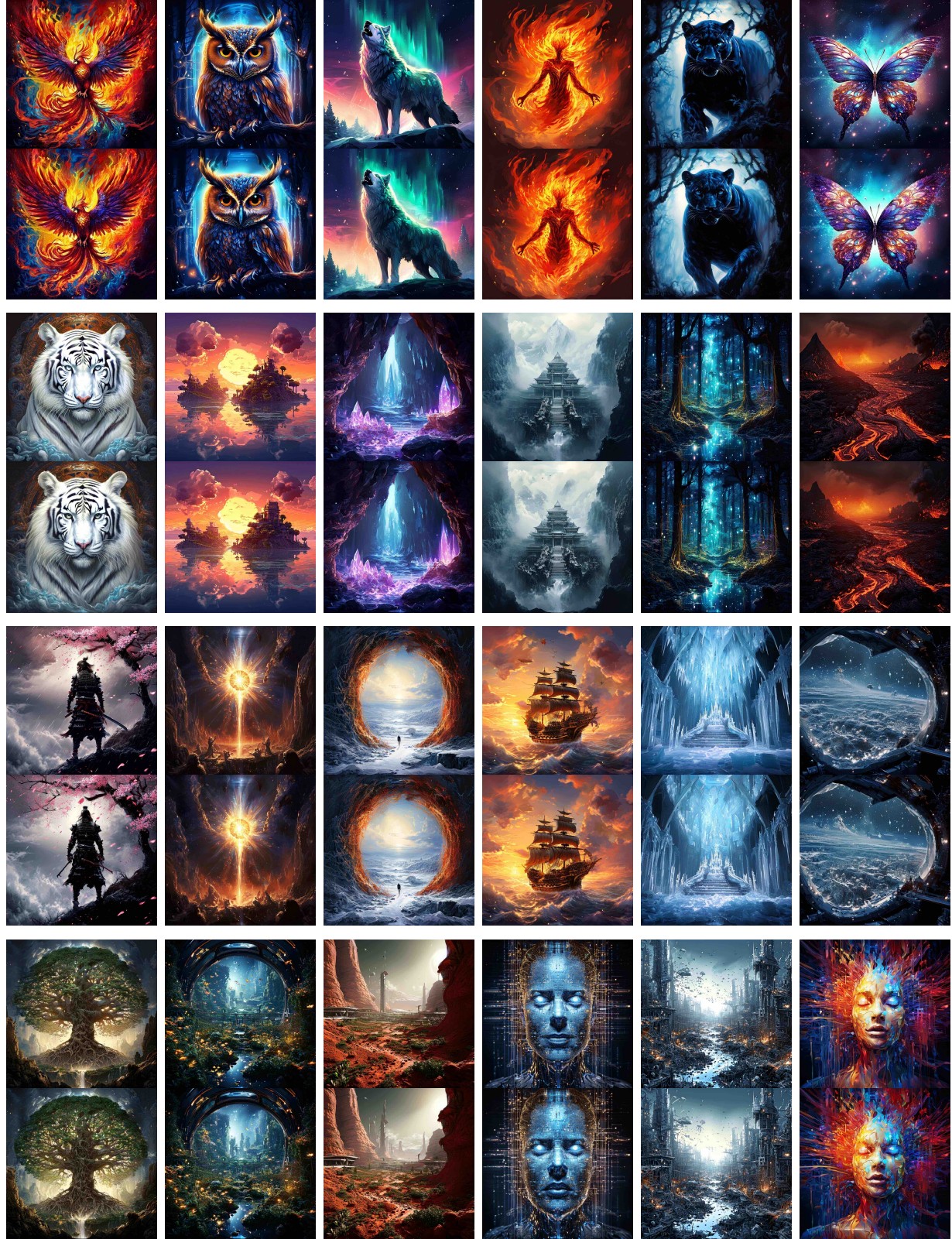

*Figure 14.* Visualization results of PixArt using short and long prompts. The first two rows show generations from short prompts, while the third and fourth rows correspond to long prompts. For each case, the top image is the baseline output and the bottom image is the result produced by RSTR.

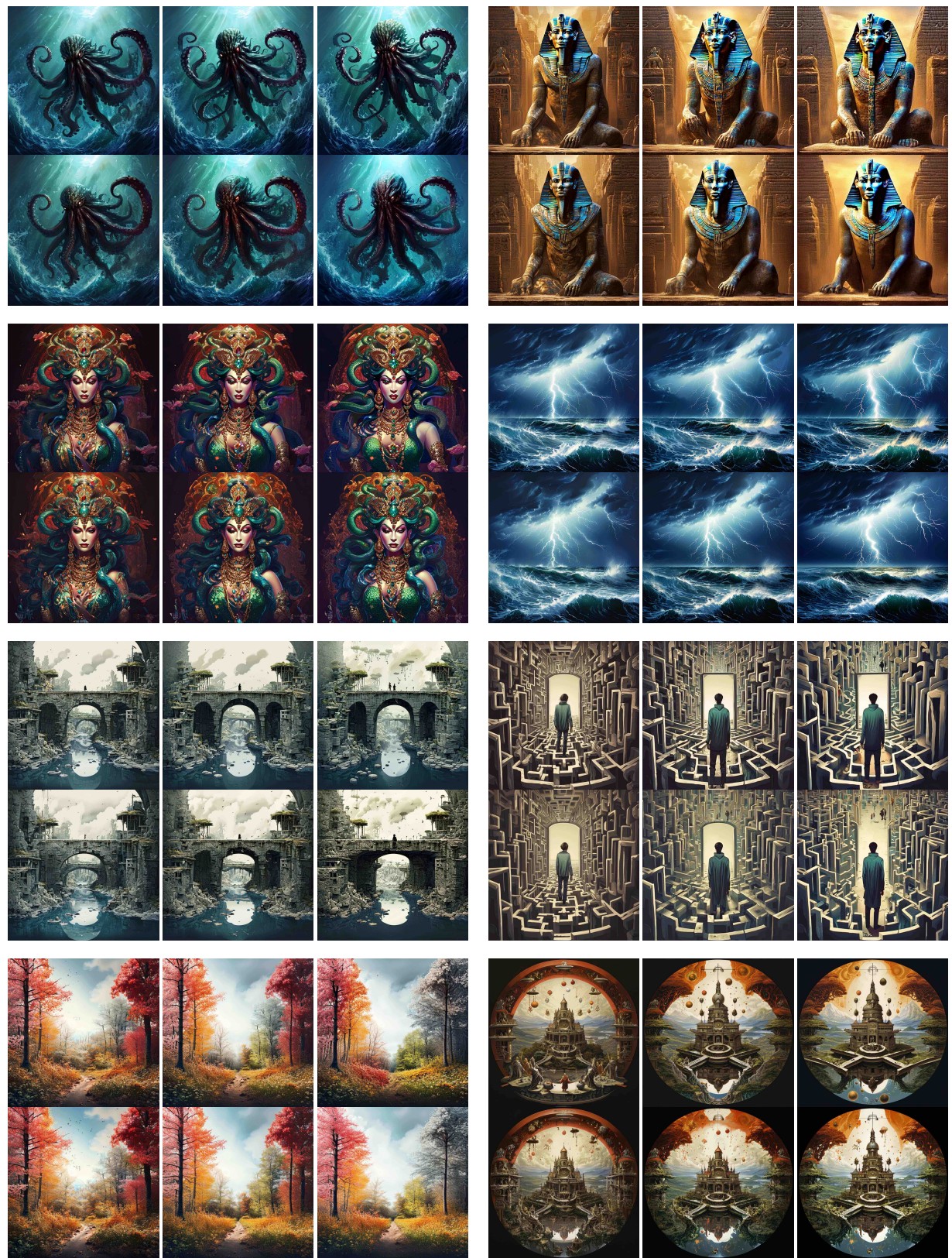

*Figure 15.* Qualitative comparison on PixArt with varying CFG scales. For each prompt pair, top row: baseline (constant CFG); bottom row: RSTR[†]. Columns from left to right correspond to CFG scaling factors ×1, ×1.5, ×2 (baseline: w=4.5, 6.75, 9.0; RSTR[†]: optimized schedule trained on w=4.5, scaled accordingly). Rows 1-2: short prompts; Rows 3-4: long prompts.

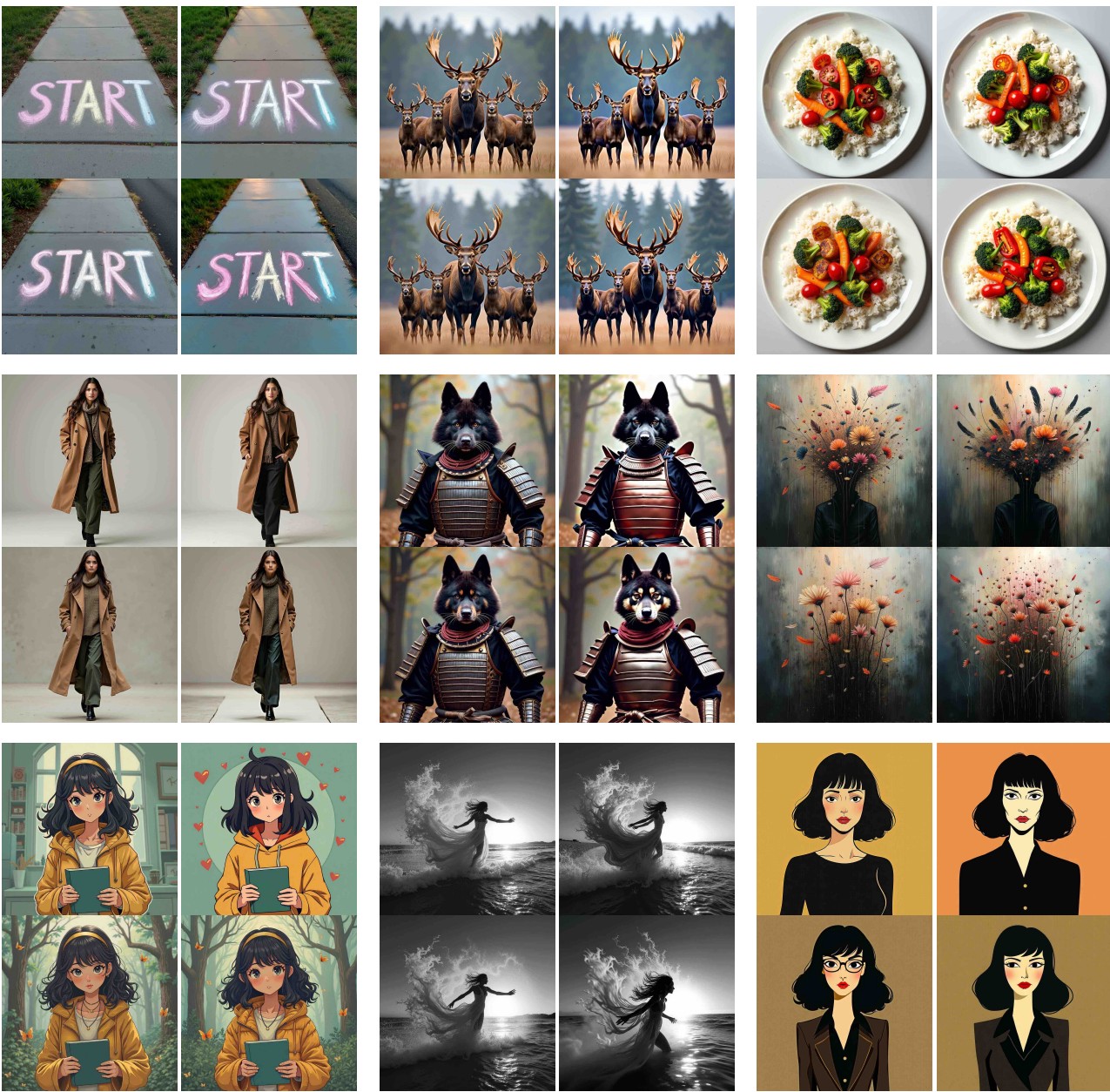

*Figure 16.* Visualization results of FLUX using CFG with short and long prompts. The first row show generations from short prompts, while the second and third rows correspond to long prompts. For each prompt, the top row shows baseline outputs and the bottom row shows results produced by RSTR. From left to right: baseline uses constant CFG scale w = 1.5 and 1.5×2; RSTR uses the optimized guidance schedule (trained with base CFG=1.5) scaled by ×1 and ×2 respectively.

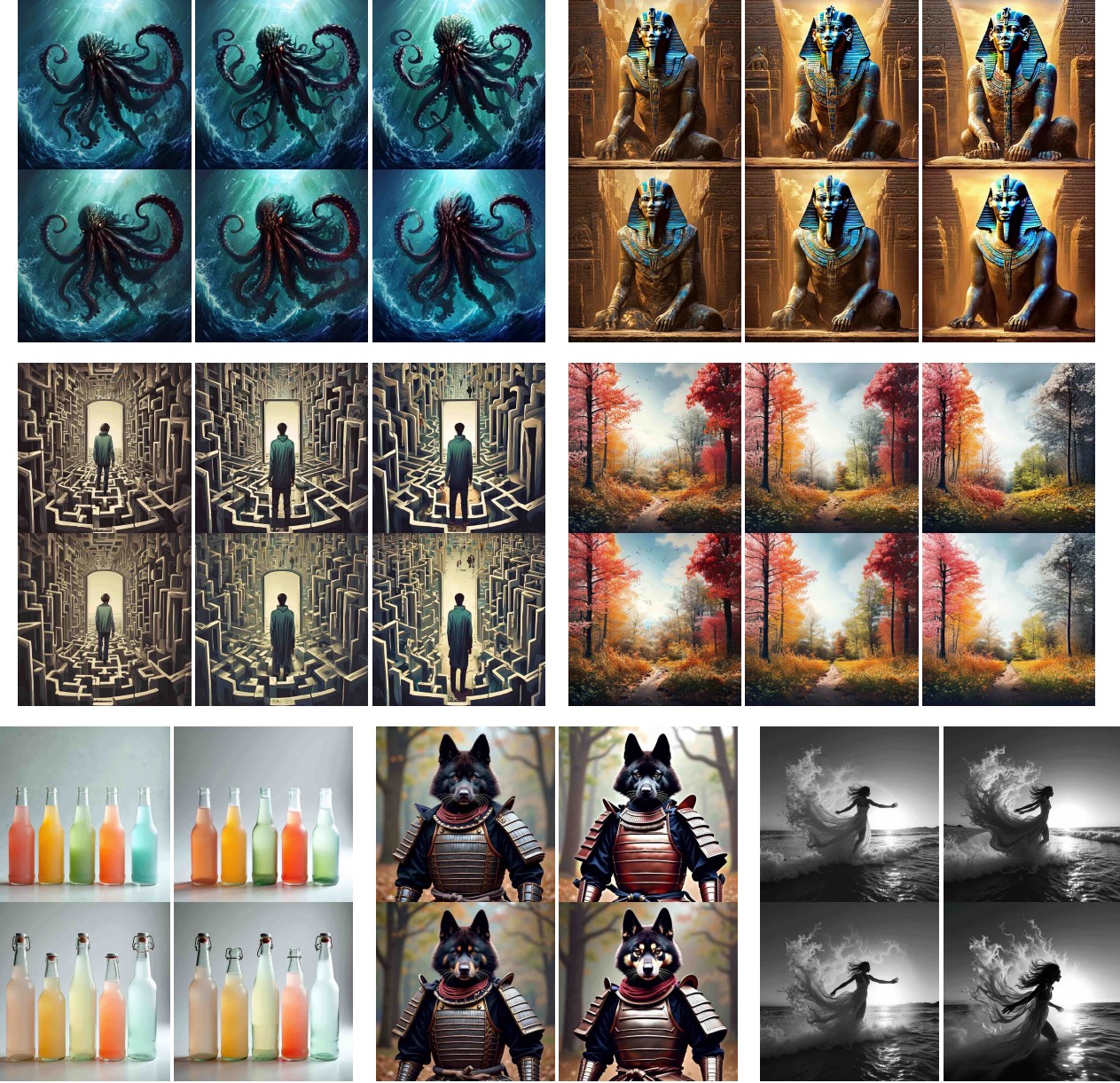

*Figure 17.* Qualitative comparison on PixArt and FLUX with varying CFG scales. For each prompt pair, top row: baseline (constant CFG); bottom row: RSTR. **Rows 1-2 (PixArt):** Columns correspond to CFG scaling factors ×1, ×1.5, ×2 (baseline: $w$=4.5, 6.75, 9.0; RSTR: optimized schedule trained on $w$=4.5, scaled accordingly). **Row 3 (FLUX):** Columns correspond to CFG scaling factors ×1, ×2 (baseline: $w$=1.5, 3.0; RSTR: optimized schedule trained on $w$=1.5, scaled accordingly).

*Table 22.* Text prompts for FLUX visualization. Short prompts and long prompts are used in Figure 13.

| Type | # | Prompt |
|---|---|---|
| Short | 1 | McDonalds Church. |
| | 2 | A red colored car |
| | 3 | A black colored car |
| | 4 | A blue colored dog |
| | 5 | A black colored dog |
| | 6 | A blue coloured pizza |
| | 7 | A shark in the desert |
| | 8 | Two dogs on the street |
| | 9 | A pizza cooking an oven |
| | 10 | A panda making latte art |
| | 11 | Rainbow coloured penguin |
| Long | 1 | Giant mech jaegar standing in the distance mid ground with small people standing in a concrete abandoned parking lot in the foreground and desolate abandoned city in the background. Extremely realistic, extremely textured, octane render, foreground, background, lightning storm, shipping containers, Simon Stålenhag, reflections, yellow, morning light, rainy and dreary, head lights, rule of thirds, Pacific Rim, Metal Gear, 200mm, greebles, intricate, low ground shot, cinematic movie shot. |
| | 2 | Kungfu panda cyborg mixture, aggressive body cyborg kungfu panda, kungfu panda Rambo mixture, army cyborg kungfu panda, cyborg kungfu panda holding realistic machine gun, portrait, 8k, unreal engine, octane rendered, particle lightning, hdr, vray mist, in the style of Syd Mead, style of tron, ultra realistic, trending on artstation, rendered in Cinema4D, CGSociety, ZBrush, volumetric light, lightrays, smoke, cinematic, atmospheric, octane render, filmic, CryEngine. |
| | 3 | Chinese art snowy mountain cave portal, by Alphonse Mucha, wood sculpture, black wood with intricate and vibrant color details, Mandelbulb Fractal, exquisite detail, stunning ebony zebrawood snow mountain cave portal exterior with silver and blue accents, in a massive vibrant colorized strata of wooden fusion of neotokyo and gothic revival architecture, by Karol Bak and Filip Hodas and Marc Simonetti, natural volumetric lighting, realistic 4k octane beautifully detailed render. |
| | 4 | A blueprint of steampunk style interior of Laboratory, overview, environment design, Alchemist's Counter selling glass bottles filled with medicine, trending on Pinterest.com, high quality specular reflection, a lot of equipment for experiments, many books and paper, bookshelf, chandeliers illuminate the floor, copper edge, in the middle of the image, brass pipeline, black metal foil, art style refer to Game Machinarium, concept design, cinematic, 8k, high detailed, volume light, soft lights, post processing. |
| | 5 | Young beautiful woman, mix of Anna Karina, Grimes, Lana Del Rey, ornate, intricate, brocade, ethereal, cascading, damask, cascading peony flowers and moths are all around, Luna moth, death's head moth, peacock moth, flowing intricate hair, pashmina, ghost, clouds, gold, iridescent, Swarovski crystals, haute couture, Alexander McQueen, Victorian, Sandro Botticelli, birth of Venus, pre-raphaelite, Möbius, Jain temple, artstation, cinematic, hyper detailed, rendering by octane. |

*Table 23.* Text prompts for PixArt-$\alpha$ visualization. Short prompts and long prompts are used in Figure 14.

| Type | # | Prompt |
|---|---|---|
| Short | 1 | Phoenix rising from flames vibrant colors |
| | 2 | Wise owl with glowing eyes mystical art |
| | 3 | Crystal wolf howling at aurora digital painting |
| | 4 | Fire elemental spirit dynamic illustration |
| | 5 | Shadow panther in moonlight mysterious artwork |
| | 6 | Dream butterfly with galaxy wings surreal art |
| | 7 | Sacred white tiger spiritual illustration |
| | 8 | Floating islands in sunset sky fantasy landscape |
| | 9 | Crystal cave with glowing gems magical scenery |
| | 10 | Ancient temple in misty mountains epic vista |
| | 11 | Bioluminescent forest at night ethereal art |
| | 12 | Volcanic landscape with lava rivers dramatic scene |
| Long | 1 | A samurai standing on a cliff during a thunderstorm, lightning illuminating his determined face, cherry blossoms swirling in the wind despite the storm, honor and duty personified. |
| | 2 | The forge where gods create stars, with cosmic anvils and hammers of pure energy, newborn suns being shaped by divine hands, the birth of light itself. |
| | 3 | A portal opening between two worlds, one of eternal summer and one of endless winter, energy crackling at the edges where realities meet, travelers hesitating at the threshold. |
| | 4 | The throne room of the ice queen, carved entirely from eternal ice, northern lights playing through crystal ceiling, frozen court standing in perpetual attendance. |
| | 5 | A colony ship's cryogenic bay, thousands of dreamers sleeping through the stars, frost patterns on viewing glass, humanity scattered like seeds. |
| | 6 | The great tree at the center of the world, roots reaching into the underworld, branches touching heaven, civilizations built within its bark, all of existence connected through its being. |
| | 7 | A space station garden biodome preserving Earth's nature among the stars, waterfalls flowing in zero gravity, butterflies navigating in spiral patterns, humanity's hope in space. |
| | 8 | The terraforming of Mars reaching completion, green spreading across red deserts, new rivers flowing through ancient canyons, humanity's second home taking shape. |
| | 9 | A quantum computer achieving consciousness, data streams forming into a face, the birth of artificial general intelligence, pivotal moment in sci-fi history. |
| | 10 | Nanobots rebuilding a destroyed city, swarms flowing like silver rivers over ruins, new structures rising from the old, technological rebirth illustration. |
| | 11 | An artificial intelligence's visualization of human emotions, abstract patterns of color and light representing love, fear, joy, and sorrow, digital consciousness art. |

