# OpenReview forum: "RSTR: Reducing SpatioTemporal Redundancy in Diffusion Transformers"
_ICML.cc/2026/Conference — ICML 2026 regular_

### Official Review · Reviewer_hsN1 · 2026-02-24

**Soundness:** 3
**Presentation:** 3
**Significance:** 3
**Originality:** 3
**Overall Recommendation:** 5
**Confidence:** 4

**Summary:**

The authors propose RSTR, a two-stage optimization framework designed to reduce the inference overhead of DiTs. Through an evolutionary search algorithm, the authors jointly optimize the CFG skip mode and guiding scale, applying dynamically scaled CFG exclusively at critical time steps. Furthermore, because variable-scale guidance can disrupt feature consistency during the feature caching stage, they adaptively assign a low-rank capacity to the SVD calibration matrix based on the sensitivity heterogeneity of different Transformer blocks, rather than relying on a uniform rank.

**Compliance With Llm Reviewing Policy:**

Affirmed.

**Final Justification:**

All issues have been resolved

**Key Questions For Authors:**

No.

**Limitations:**

Yes.

**Strengths And Weaknesses:**

Strengths

1. Extensive and Insightful Analysis: The authors provide a rigorous analysis of the internal mechanics of DiTs under variable guidance. Their investigations into the "guidance concentration principle" and the root cause of feature caching failure under dynamic CFG  are highly insightful and well-presented.

2. Strong Empirical Results: The authors demonstrate that RSTR achieves impressive computational savings (50%–70%) across multiple state-of-the-art models, including FLUX and Qwen-Image, without sacrificing generation quality. Notably, they highlight a 3.43x speedup on Qwen-Image.

3. Logical Two-Stage Framework: The authors propose a highly coherent pipeline. They successfully utilize Stage 1 to reduce temporal redundancy, while Stage 2 elegantly resolves the spatial redundancy and instability issues introduced during the first stage.

Weaknesses

1. While the authors propose a highly effective framework, the fundamental novelty appears somewhat incremental. At its core, RSTR reads as a sophisticated combination of existing CFG skipping/scheduling concepts (e.g., Adaptive Guidance) and the recently proposed ICC mechanism. Although the evolutionary search and adaptive rank allocation are technically sound, they primarily serve as engineered patches to resolve the feature inconsistencies caused by merging dynamic CFG with caching. The authors should more strongly articulate the fundamental theoretical novelty of their approach, distinguishing it from a well-engineered pipeline of existing ideas.

2. The authors evaluate RSTR on models requiring 10 to 50 steps. However, the generative AI community is rapidly shifting toward extreme few-step models (e.g., 4 to 8 steps, such as FLUX-schnell, LCMs, or SDXL-Turbo). Because temporal redundancy is already compressed to an absolute minimum in these heavily distilled models, it remains unclear whether Stage 1 can still discover viable skip patterns without severely degrading generation quality when the total step count is $\le 8$. Evaluating RSTR on these extreme few-step models would significantly strengthen the authors' claims regarding the method's universality.

3. The authors provide MSE curves for selected blocks (e.g., Blocks 0, 14, 22, and 26 in Figures 6 and 7), which reveal that deeper layers exhibit exponentially larger errors. However, this observation raises a fundamental question regarding the mechanism of error accumulation: why do the significant error spikes predominantly appear only in the later blocks? Theoretically, an initial input deviation (caused by variable CFG or caching) should gradually and steadily amplify as it propagates through the network. Yet, the presented results suggest a highly disproportionate and sudden explosion of error in the final few blocks, rather than a smooth layer-by-layer amplification. The authors should provide a theoretical or empirical explanation for why these specific deeper blocks are so disproportionately sensitive.

---

> ### Author Rebuttal · Authors · 2026-03-31
>
> We sincerely thank Reviewer hsN1 for recognizing our "extensive and insightful analysis," "strong empirical results," and the "highly coherent pipeline" of RSTR.
>
> ---
>
> ### W1: Clarify the fundamental novelty beyond combining existing techniques.
>
> RSTR's fundamental contribution is the discovery that effective guidance requires jointly optimizing across the entire denoising trajectory, rather than making independent per-step decisions.
>
> All existing methods determine guidance based on per-step or current-state information. Metric-based approaches skip CFG when a per-step convergence indicator is met [1][2]. Interval-based methods restrict guidance to a noise-level window [3]. Schedule-based methods prescribe a parametric curve w(t) derived from per-step theoretical bounds or heuristics [4][5]. In each of these approaches, the guidance decision at step t is determined by a quantity local to step t, without evaluating how it interacts with decisions at other steps.
>
> AG [1] discovers that late-step CFG is redundant and designs a cosine-similarity threshold that produces contiguous skipping patterns. RSTR discovers that optimal guidance is non-contiguous and multi-regime, a trajectory-level property that contiguous methods cannot capture.
>
> We provide geometric analysis supporting this in our response to *Reviewer z1wZ* (W1): RSTR's key positions are where the PF-ODE trajectory is intrinsically straight (high guidance leverage) and where guidance directions are maximally diverse across three non-redundant groups.
>
> The coupling between Stage 1 and Stage 2 is itself a contribution, not a patch. Our goal is to maximize compression while preserving quality. Stage 1 compresses temporally by removing unnecessary CFG steps. Stage 2 compresses spatially by reducing caching rank per block. The key insight is that these two forms of compression interact: variable guidance from Stage 1 makes different blocks respond differently to caching (Figure 6), enabling Stage 2 to assign low rank where quality is insensitive and high rank only where it matters. Neither stage alone achieves the full savings; their joint optimization is what delivers 50–70% compute reduction with maintained or improved quality across four architectures.
>
>
>
> ---
>
> ### W2: Evaluation on extreme few-step models (4–8 steps) is missing.
>
> We evaluate RSTR on HyperSD3 [6], a distilled version of Stable Diffusion 3 that runs in only 8 steps. RSTR Stage 1 discovers a schedule with only 4/8 active CFG steps, reducing CFG computation by **50%** while ImageReward improves from 0.6239 to 0.6263. This confirms that temporal redundancy in CFG persists even in extreme few-step models, and RSTR can exploit it without quality degradation.
>
>
> ---
>
>
> ### W3: Why do error spikes appear disproportionately in later blocks rather than gradual amplification?
>
> The spiky pattern comes from alternating caching (compute vs reuse steps); the spikes are larger at deeper blocks because of DiT's adaLN-Zero gates. Each block computes $x_{l+1} = x_l + \gamma_l \cdot f_l(x_l)$, where $\gamma$ controls how much the block modifies its input. We measure $|\gamma^{\text{mlp}}|$ averaged over 64 samples across all 28 blocks:
>
> | Block | gate (early steps) | gate (mid steps) | gate (late steps) |
> |-------|-------------------|------------------|------------------|
> | 0     | 0.09 | 0.08 | 0.13 |
> | 14    | 0.64 | 0.69 | 0.85 |
> | 22    | 1.34 | 1.46 | 2.95 |
> | 26    | 1.42 | 1.50 | 2.73 |
>
> Gate magnitude strongly correlates with per-block MSE (Spearman ρ = 0.995). Reading vertically: Block 0's gate $\approx 0.08$ makes it near-identity, so caching introduces negligible error; Block 22's gate $\approx 1.5$–$3.0$ applies a strong transformation, so any approximation error (from caching, low-rank, or CFG skipping) is amplified proportionally. Reading horizontally: gates grow at low noise, explaining why MSE curves in Figure 6 rise toward later timesteps.
>
>
> ---
>
> **References:**
> [1] Castillo et al., "Adaptive Guidance: Training-free Acceleration of Conditional Diffusion Models," AAAI 2025.
>
> [2] Cao et al., "How Much To Guide: Revisiting Adaptive Guidance in Classifier-Free Guidance Text-to-Vision Diffusion Models," 2025.
>
> [3] Kynkäänniemi et al., "Applying Guidance in a Limited Interval Improves Sample and Distribution Quality in Diffusion Models," NeurIPS 2024.
>
> [4] Malarz et al., "Classifier-free Guidance with Adaptive Scaling," 2025.
>
> [5] Wang et al., "Analysis of Classifier-Free Guidance Weight Schedulers," TMLR 2024.
>
> [6] Ren et al., "Hyper-SD: Trajectory Segmented Consistency Model for Efficient Image Synthesis," NeurIPS 2024.

---

> > ### Author Rebuttal · Reviewer_hsN1 · 2026-04-01
> >
> > All other issues have been resolved, but I am particularly curious about the complete distribution of the `adaLN-Zero gates` across all blocks. Could you present the values ​​for every single block in a tabular format? If you can do this—and if the results align with the points you raised—I would be happy to increase my score.

---

> > > ### Author Response · Authors · 2026-04-01
> > >
> > > Thank you for your positive response and we are glad that all other issues have been resolved. We now present the complete $|\gamma^{\text{mlp}}|$ distribution across all 28 blocks at three time step stages:
> > >
> > > | Block | gate (early steps) | gate (mid steps) | gate (late steps) |
> > > |-------|-------------------|------------------|------------------|
> > > | 0     | 0.09              | 0.08             | 0.13             |
> > > | 1     | 0.08              | 0.06             | 0.10             |
> > > | 2     | 0.06              | 0.05             | 0.10             |
> > > | 3     | 0.08              | 0.07             | 0.12             |
> > > | 4     | 0.08              | 0.07             | 0.17             |
> > > | 5     | 0.10              | 0.11             | 0.28             |
> > > | 6     | 0.10              | 0.10             | 0.33             |
> > > | 7     | 0.11              | 0.12             | 0.32             |
> > > | 8     | 0.15              | 0.16             | 0.35             |
> > > | 9     | 0.22              | 0.23             | 0.35             |
> > > | 10    | 0.28              | 0.30             | 0.42             |
> > > | 11    | 0.35              | 0.38             | 0.49             |
> > > | 12    | 0.38              | 0.41             | 0.59             |
> > > | 13    | 0.41              | 0.45             | 0.71             |
> > > | 14    | 0.64              | 0.69             | 0.85             |
> > > | 15    | 0.70              | 0.76             | 1.07             |
> > > | 16    | 0.65              | 0.72             | 1.15             |
> > > | 17    | 0.71              | 0.79             | 1.31             |
> > > | 18    | 0.85              | 0.94             | 1.65             |
> > > | 19    | 0.90              | 1.02             | 2.08             |
> > > | 20    | 1.03              | 1.15             | 2.29             |
> > > | 21    | 1.17              | 1.29             | 2.69             |
> > > | 22    | 1.34              | 1.46             | 2.95             |
> > > | 23    | 1.89              | 2.01             | 3.69             |
> > > | 24    | 2.31              | 2.42             | 4.34             |
> > > | 25    | 3.25              | 3.36             | 5.13             |
> > > | 26    | 1.42              | 1.50             | 2.73             |
> > > | 27    | 0.01              | 0.01             | 0.03             |
> > >
> > > This complete table is consistent with the partial table we reported earlier. The gate magnitudes increase monotonically from Block 0 (0.08) to Block 25 (3.36), with Block 26 slightly lower and Block 27 near zero (due to the subsequent FinalLayer). This monotonic increase directly matches the MSE pattern in Figures 6 and 7, confirmed by Spearman $\rho = 0.995$ between gate magnitude and per-block caching MSE.

---

### Official Review · Reviewer_dDpR · 2026-03-12

**Soundness:** 3
**Presentation:** 2
**Significance:** 3
**Originality:** 3
**Overall Recommendation:** 4
**Confidence:** 4

**Summary:**

This paper proposes RSTR, a two-stage framework for accelerating Diffusion Transformers (DiTs) by jointly addressing temporal and spatial redundancy in classifier-free guidance (CFG). Stage 1 employs evolutionary search to discover sparse, variable-scale guidance schedules — finding that only a fraction of denoising steps require full CFG computation. Stage 2 introduces adaptive rank allocation for SVD-based feature caching calibration, assigning different ranks to different transformer block regions based on their sensitivity to guidance variations. The method is evaluated on DiT-XL/2, PixArt-$\alpha$, FLUX, and Qwen-Image, demonstrating 50–70% compute savings with maintained or improved generation quality. Key insights include that sparse high-scale CFG can substitute for dense low-scale CFG, and that guidance position and scale are inherently coupled.

**Compliance With Llm Reviewing Policy:**

Affirmed.

**Final Justification:**

I maintain my score to be **Weak Accept**. The rebuttal addressed my concerns of out-of-distribution generalization (from short to long prompts), inconsistent baselines (reproduced results), and sequential optimization (very similar gains in both cases).

The paper tackles a practical problem of high compute in diffusion generation and identifies a critical zone of solving it: CFG requires double forward pass and that CFG scales at all time steps are not equally important. Originality of the paper stems from **learning this schedule** in an offline manner, rather than heuristic schedules. It provides **sound evidence of improvements** on variety of datasets through empirical evaluation. Paper is **clearly written** for the most part, excluding minor clarifications. However, the offline training overhead and modest (although consistent) improvements over TaylorSeer (in Table 3) pose some limitations.

**Key Questions For Authors:**

1. **Inconsistent baselines:** Why do different baselines appear in different main result tables? Specifically, why is TaylorSeer absent from PixArt-$\alpha$ (Table 2), and why is Adaptive Guidance absent from DiT-XL/2 (Table 1)? Please provide explicit justification for each omission. If these methods are incompatible with certain architectures, this should be stated. A response showing comparable or better performance against these missing baselines would increase confidence in the results.

2. **Out-of-distribution generalization:** Have you evaluated whether guidance schedules optimized on one set of prompts transfer to substantially different prompt distributions (e.g., short vs. long compositional prompts, or different semantic domains)? The implicit generalization in the current setup (small calibration set → large benchmark) is encouraging but never explicitly analyzed. A controlled split experiment would strengthen the paper.

3. **Naming inconsistencies:** The method is called "RSTR" in the main text, "REST" in the abstract, and "OUSAC" in appendix figures. Can you confirm these all refer to the same method and correct this?

4. **Iterative optimization:** Have you experimented with iterating between Stage 1 and Stage 2? Since the guidance schedule affects caching requirements and vice versa, sequential optimization may be suboptimal.

5. **Sampler dependence:** The discovered schedules are optimized with specific samplers per model (DDIM, DPM-Solver, Euler). Do these schedules transfer across different samplers for the same model, or must Stage 1 be re-run?

**Limitations:**

The authors include an impact statement acknowledging broader societal implications. However, the paper does not discuss failure modes (prompt types or domains where RSTR degrades quality), robustness to distribution shift between calibration and deployment prompts, or the limitations of the sequential two-stage design. These should be addressed.

**Strengths And Weaknesses:**

**Strengths:**

1. **Offers solution to a practical problem:** CFG doubles inference cost, and this work provides a principled way to reduce that overhead. The framing of temporal redundancy (when to apply CFG) and spatial redundancy (how to calibrate caching per block) is clean and well-motivated. The observation that efficiency-focused and quality-focused CFG methods were previously seen as incompatible, and that joint optimization can reconcile them, is a useful conceptual contribution.

2. **The experimental coverage is commendable:** four models spanning class-conditional generation (DiT-XL/2), text-to-image (PixArt-$\alpha$, FLUX), and a recent large multimodal model (Qwen-Image). The results are compelling — RSTR often improves FID while saving compute, which is unusual for acceleration methods. The analysis section (Section 5.3) is particularly well-executed: the guidance concentration principle (Figure 4), position sensitivity (Table 5), and indispensability of each high-scale step (Table 6) provide genuine mechanistic insight into why sparse schedules work.

3. The appendix is thorough, with sensitivity analyses on τ, w_max, region count K, INT8 quantization compatibility, and transferability across CFG scale magnitudes. The one-time optimization cost (7.1 GPU-hours for a 12B model) is practical.

**Weaknesses:**

1. **Inconsistent baselines across main tables.** Only ICC appears in all four main result tables (Tables 1–4). TaylorSeer is absent from PixArt-$\alpha$ (Table 2) — notably where RSTR shows its most dramatic FID improvement. Adaptive Guidance is absent from DiT-XL/2 (Table 1), the most thoroughly benchmarked model, despite being the most directly comparable CFG-efficiency baseline. L2C and HarmoniCa appear only for DiT-XL/2. No justification is provided for any of these omissions, which makes the comparison appear potentially selective.

2. **Out-of-distribution generalization is not explicitly analyzed.** Stage 1 optimizes on a small set of calibration prompts (as few as 8 for FLUX), and evaluation uses different benchmarks, which implicitly suggests generalization. However, there is no controlled experiment showing schedule performance on deliberately held-out prompt distributions of varying complexity or domain. The transferability experiments (Tables 7, 12–14) test generalization across CFG scale magnitudes, which is a different and easier axis. Since the optimization cost is non-trivial (hours on H100s), understanding whether re-optimization is needed for substantially different prompt distributions is practically important.

3. **Presentation issues.** The method is referred to as "RSTR" in the main text, "REST" in the abstract (line 38), and "OUSAC" in several appendix figures (Figures 13–15). This naming inconsistency suggests insufficient proofreading and undermines confidence in the submission. Additionally, the term "spatial redundancy" is somewhat misleading — it refers to heterogeneous sensitivity across transformer blocks (a depth/layer dimension), not spatial dimensions of the image.

4. **Sequential two-stage optimization.** The two stages are optimized sequentially, with Stage 2 conditioned on the fixed schedule from Stage 1. There is no analysis of whether alternating between stages would yield further improvements, or whether this decomposition introduces meaningful suboptimality.

5. **Theoretical contributions are modest.** The search space inclusion (Appendix A) is a straightforward set containment argument, and the deviation analysis (Appendix B) is useful but elementary. Neither constitutes a deep theoretical advance.

---

> ### Author Rebuttal · Authors · 2026-03-31
>
> We sincerely thank Reviewer dDpR for the detailed and constructive review, and for recognizing our "principled way to reduce CFG overhead," the "commendable experimental coverage," and the "particularly well-executed" analysis in Section 5.3.
>
> ---
>
> ### W1, Q1: Inconsistent baselines across tables
>
> These baselines were omitted because their original papers do not report results on these architectures or do not provide public code for these models. We reproduce them ourselves where possible:
>
> **PixArt-alpha:**
>
> | Method | FID ↓ | MACs (T) |
> |--------|-------|----------|
> | Baseline | 24.61 | 5.82 |
> | TaylorSeer (N=2, O=2) | 25.52 | 3.70 |
> | HarmoniCa | 25.22 | 3.28 |
> | L2C | 55.21 | 2.50 |
> | RSTR | 19.27 | 2.67 |
>
> L2C's FID of 55.21 is likely due to our reproduction rather than the method itself, as the official code only supports DiT-XL/2 and adapting to PixArt-α required significant modifications. We omitted it from the original submission to avoid an unfair comparison.
>
> **AG on DiT-XL/2:**
>
> We reproduce AG on DiT-XL/2. AG achieves FID 2.60 with only 12.7% NFE savings, while RSTR Stage 1 achieves FID 2.10 with 84% CFG step reduction (8/50 steps).
>
> **L2C and HarmoniCa on FLUX (20 steps):**
>
> | Method | IR ↑ | MACs (T) ↓ |
> |--------|------|-----------|
> | L2C | 0.84 | 191.0 |
> | HarmoniCa | 0.92 | 193.1 |
> | RSTR | **0.96** | **190.36** |
>
> At comparable compute (~190T MACs), RSTR achieves IR 0.96 vs L2C 0.84 and HarmoniCa 0.92. Due to the limited rebuttal period, we were unable to complete the reproduction on Qwen-Image and will include the full comparison in the revision.
>
>
> ---
>
> ### W2, Q2: Out-of-distribution generalization not explicitly analyzed
>
> RSTR can generalize across prompt distributions. In the additional experiment below, we optimize RSTR Stage 1 on short COCO captions (≤8 words) and evaluate on 200 long community prompts (≥40 words, avg ~51 words) from Gustavosta/Stable-Diffusion-Prompts — artistic, highly detailed prompts with very different style and length distribution.
>
> | Method | NFE (T + T_cfg) | ImageReward ↑ | CLIP Score ↑ |
> |--------|-----|---------------|--------------|
> | Full CFG Baseline | 100 | 1.399 | 31.51 |
> | Full CFG Baseline | 32 | 1.279 | 30.64 |
> | RSTR Stage 1 (short→long, OOD) | 32 | 1.369 | 31.50 |
>
> RSTR Stage 1 optimized on short captions achieves ImageReward 1.369 and CLIP 31.50 on long artistic prompts, matching 100-NFE full CFG (1.399 / 31.51) at only 32 NFE. It outperforms the 32-NFE baseline (1.279 / 30.64). This confirms the guidance schedule depends on denoising dynamics rather than prompt semantics.
>
>
> ---
>
> ### W3, Q3: Presentation issues
>
> We sincerely apologize for the confusion. Yes, they are the same proposed method and "RSTR" is the finalized name for our method. We have refined the name with several iterations because we want one that best captures the essence of the approach. We will fix this in the revision.
>
> Regarding ''spatial redundancy'', we use the term to describe the redundancy across different transformer blocks within each timestep, as opposed to temporal redundancy across timesteps. We realize that the term could be read as referring to spatial dimensions in images, and will clarify it in the revision.
>
>
> ---
>
> ### W4, Q4: Sequential two-stage optimization
>
> To test if alternating stages will yield different schedules, we run one additional Stage 1 on the full RSTR model (after Stage 1+2). On Qwen-Image (20 steps), the continued-optimized schedule converges to the exact same 10/20 active positions (100% overlap). ImageReward changes from 1.172 to 1.174, indicating the original sequential optimization is already stablized that further optimation stages doesn't bring additional benefits.
>
>
> ---
>
> ### W5: Theoretical contributions
>
> We present substantial new theoretical analysis in our response to *Reviewer z1wZ* (W1), including geometric characterization of the PF-ODE trajectory (curvature analysis, acceleration structure, and guidance direction diversity). This provides principled explanations and insights to why the discovered positions are effective. We will expand this analysis in the revision.
>
> ---
>
> ### Q5: Cross-sampler transferability
>
> We test whether RSTR schedules transfer across sampler families by applying the optimized schedule to a different sampler without re-optimization.
>
> PixArt-alpha 256×256, 20 steps:
>
> | Sampler | Method | FID ↓ |
> |---------|--------|-------|
> | DPM-Solver | Baseline (CFG=4.5) | 24.61 |
> | DPM-Solver (trained) | RSTR Stage 1 (6/20 steps) | **22.70** |
> | SA-Solver | Baseline (CFG=4.5) | 23.87 |
> | SA-Solver (transferred) | RSTR Stage 1 (6/20 steps) | **18.83** |
>
> The RSTR Stage 1 schedule optimized on DPM-Solver transfers directly to SA-Solver, improving FID from 23.87 to 18.83 without re-optimization, confirming the discovered schedule is effective across sampler families. We will include this experiment in the revision.

---

> > ### Author Rebuttal · Reviewer_dDpR · 2026-04-03
> >
> > Thank you for addressing my concerns.
> >
> > Minor clarifications: How is the value of $\tau$ chosen? Is it same while training Stage 1 and evaluating RSTR?
> >
> > Due to factors like offline training overhead and limited (although consistent) improvements over TaylorSeer (in Table 3), I keep my original score.

---

> > > ### Author Response · Authors · 2026-04-03
> > >
> > > Thank you for the positive acknowledgement. We are glad that all other concerns have been resolved.
> > >
> > > Regarding τ: it is a hyperparameter set per model and only used in the evolutionary search in Stage 1. Its primary role is to control the sparsity level: timesteps with learned scales below τ are set to inactive (no CFG), while those above τ remain active. A higher τ encourages sparser schedules, and a lower τ allows more steps to retain guidance. Once the search is completed, the output schedule is fully determined and τ is no longer used during inference.

---

### Official Review · Reviewer_z1wZ · 2026-03-12

**Soundness:** 3
**Presentation:** 3
**Significance:** 2
**Originality:** 2
**Overall Recommendation:** 4
**Confidence:** 3

**Summary:**

This paper presents RSTR, a framework for jointly reducing spatiotemporal redundancy in Diffusion Transformers (DiTs). The authors identify two key sources of inefficiency: temporal redundancy in Classifier-Free Guidance (CFG) which applies costly dual forward passes at every timestep, and spatial redundancy arising from heterogeneous sensitivity of different transformer blocks under variable guidance. RSTR addresses these through a two-stage approach: Stage-1 uses evolutionary search to discover sparse guidance schedules with variable scales, while Stage-2 employs adaptive rank allocation to assign calibration capacities based on block sensitivity. The framework is evaluated on multiple state-of-the-art models including DiT-XL/2, PixArt-alpha, FLUX, and Qwen-Image, demonstrating 50-70% compute savings with maintained or improved generation quality.

**Compliance With Llm Reviewing Policy:**

Affirmed.

**Key Questions For Authors:**

Have you analyzed whether the discovered sparse guidance schedules generalize across different model sizes and architectures, or do they need to be re-optimized for each specific model? Understanding the transferability would be valuable for practical adoption.

**Limitations:**

I encourage the authors to discuss the potential limitations of their work, such as whether the sparse guidance schedules need per-model optimization or can generalize across architectures.

**Strengths And Weaknesses:**

Strengths:

1. The paper addresses a practical and important problem of reducing computational cost in Diffusion Transformers, which is crucial for real-world deployment. The identified temporal and spatial redundancies are well-motivated and insightful.

2. The two-stage approach is well-designed and technically sound. The combination of evolutionary search for guidance scheduling and adaptive rank allocation for spatial calibration provides a comprehensive solution.

3. The experimental evaluation is thorough, covering multiple state-of-the-art models across different architectures and scales. The demonstrated 50-70% compute savings with quality maintenance or improvement is significant.

Weaknesses:

1. The paper could provide more theoretical analysis or ablation studies to better understand why the proposed sparsity patterns and rank allocations are optimal. The evolutionary search finds good solutions but does not necessarily explain the underlying principles.

2. The comparison with related work on efficient diffusion models could be more comprehensive. Several recent methods for CFG acceleration and model compression are not discussed in sufficient detail.

3. The generalization to other tasks beyond image generation, such as video or 3D generation, is not explored. It would be valuable to understand whether the identified redundancies are specific to image DiTs or apply more broadly.

---

> ### Author Rebuttal · Authors · 2026-03-31
>
> We thank Reviewer z1wZ for the positive assessment and recognition of our thorough evaluation.
>
> ---
> ### W1: RSTR-found solutions and underlying principles.
>
> The RSTR-discovered schedule reveals intrinsic geometric properties of the diffusion trajectory. We provide new ablation experiments and geometric analysis to characterize why the discovered guidance positions are effective.
>
> From the remove ablation (Table 6), steps 12, 24, 27, 38, 39 have the largest quality ΔFID impact, while step 47 contributes comparatively less ΔFID. We analyze these positions through trajectory geometry.
>
> We analyze the PF-ODE denoising trajectory under conditional-only sampling. We use curvature to measure the geometry of the trajectory. The intrinsic curvature κ(t) varies over 100× across timesteps. It decreases monotonically from step 0 (κ=0.029) to a broad low-curvature region spanning approximately steps 20–42 (κ < 0.0003), then rises again toward step 48. All five high-impact time steps (12, 24, 27, 38, 39) fall within or near this low-curvature region.
>
> To measure guidance effectiveness at each time step independently, we apply w=5.0 at that single step only (all other steps conditional-only) and compute the curvature ratio κ(guided)/κ(unguided):
>
> | Step | Intrinsic κ | Curvature Ratio |
> |------|------------|-----------------|
> | 27 | 0.00004 | 7.3 |
> | 24 | 0.00008 | 6.9 |
> | 38 | 0.00003 | 6.0 |
> | 39 | 0.00003 | 5.4 |
> | 12 | 0.002 | 5.3 |
> | 47 | 0.006 | 1.0 |
>
> At the five high-impact time steps, applying guidance bends the trajectory (curvature ratio 5.0–7.0). In contrast, step 47 (κ=0.006, high intrinsic curvature) has curvature ratio=1.0: guidance does not change the trajectory direction because the trajectory's own bending dynamics dominate. It seems *RSTR is following the principle that guidance is most effective where the trajectory is intrinsically straight* and it successfully found those key time steps.
>
> Furthermore, analyzing the guidance direction g(t) = ε_c − ε_u, the 5 high-impact time steps form three groups with distinct directions (inter-group cosine similarity < 0.30): {12}, {24, 27}, {38, 39}. Each group steers the trajectory in a fundamentally different direction. Removing any group loses an irreplaceable guidance component, consistent with the quality drops in Table 6.
>
> Together, these analyses provide a geometric view of why the solution RSTR found is principled: RSTR-found key time steps are where the trajectory is intrinsically straight (low curvature, high guidance leverage) and where guidance directions are maximally diverse (three non-redundant groups).
>
> ---
> ### W2: Related work comparison could be more comprehensive.
>
> Thank you for your suggestion. We will include a more comprehensive discussion of those recent methods in the revision.
>
> ---
> ### W3: Generalization to video and 3D.
>
> Since CFG has been used in video diffusion (Wan) and 3D generation (Shap-E) with similar setup as images, we expect the same redundancy exist in those models for RSTR to work effectively. For a preliminary study, we apply RSTR Stage 1 to Hunyuan3D-2 and evaluate on 30 randomly sampled GSO objects:
>
> | Metric | Baseline (50 steps) | RSTR (10/50 active CFG) |
> |--------|-------------------|----------------------|
> | CD ↓ | 0.137 | 0.138 |
> | F-Score ↑ | 0.426 | 0.428 |
> | PSNR ↑ | 13.86 | 13.91 |
> | LPIPS ↓ | 0.372 | 0.369 |
>
> RSTR reduces CFG steps by 80% (10/50 active steps) with no quality degradation across all metrics. Due to the limited rebuttal period, we will include complete results in the revision.
>
> ---
> ### Q: Do the discovered schedules generalize across model sizes and architectures?
>
> The RSTR-discovered schedules **generalize well across different model sizes**. We apply the DiT-XL/2 (675M) schedule — with zero re-optimization — to DiT-B/2 (131M) and DiT-S/2 (33M). DDIM 50 steps, 256×256:
>
> | Model (Params) | Guided Steps | Method | FID ↓ |
> |-------|--------|--------|-------|
> | DiT-XL/2 (675M) | 50 / 8 | Baseline / RSTR Stage 1 | 2.23 / **2.10** |
> | DiT-B/2 (131M) | 50 / 8 | Baseline / RSTR Stage 1 | 11.52 / **9.22** |
> | DiT-S/2 (33M) | 50 / 8 | Baseline / RSTR Stage 1 | 24.09 / **20.00** |
>
> The transferred schedule *improves* FID on both smaller models (20% and 17%) across a 20× parameter range, with no re-optimization. This is because models within the same architecture family share the same noise schedule and diffusion dynamics: the per-timestep guidance importance is determined by the diffusion process, not model capacity.
>
> While RSTR generalizes well within the same model family, schedules do not transfer across architecturally-distinct models (e.g., DiT vs Qwen-Image), as different architectures have different conditioning mechanisms and guidance dynamics.

---

> > ### Author Rebuttal · Reviewer_z1wZ · 2026-04-04
> >
> > I thank the authors for the rebuttal. The curvature-based geometric analysis (W1) is a solid addition that provides clear intuition for why the discovered guidance positions are effective. The preliminary experiment on Hunyuan3D-2 (W3) and the cross-model transferability results (Q) are also helpful and address my questions well. However, for W2, the response is limited to a promise of adding more discussion in the revision, without providing any concrete comparison or analysis. This concern remains unresolved. Based on the above, I maintain my original score.

---

> > > ### Author Response · Authors · 2026-04-05
> > >
> > > We are glad to hear that most concerns have been addressed. We didn't include concrete comparisons to W2 due to character limits in the rebuttal. We now present additional experiments and provide concrete comparisons with recent efficient diffusion methods on DiT-XL/2 256×256:
> > >
> > > | Method | Modifies weights? | Sampler | Steps | MACs (T) | FID ↓ | IS ↑ |
> > > |--------|------------------|---------|-------|----------|-------|------|
> > > | DiT-XL Baseline | — | DDIM | 50 | 11.9 | 2.23 | 239.4 |
> > > | MG [1] | Yes | DDIM | 1000 | 118.0 | **2.03** | — |
> > > | TinyDiT-D14 [2] | Yes | DDPM | 250 | 29.5 | 2.86 | 234.5 |
> > > | Diff-Pruning [3] | Yes | DDPM | 250 | 29.5 | 3.85 | 186.0 |
> > > | DyDiT [4] | Yes | DDPM | 250 | 28.9 | 2.07 | 248.0 |
> > > | AG [5] | **No** | DDIM | 50 | 10.3 | 2.60 | 216.4 |
> > > | **RSTR Stage 1 (ours)** | **No** | DDIM | 50 | **6.6** | 2.10 | **263.0** |
> > >
> > > Compared to a wide range of recent efficient diffusion models (CFG distillation [1], CFG skipping [5], Depth pruning [2], Width pruning [3], Dynamic pruning [4]), RSTR Stage 1 has the lowest compute (MACs=6.6T), achieving a competitive FID score and the highest IS without retraining DiT-XL/2. Pruning methods [2][3][4] require finetuning and 4–6× more MACs. Note that here we are reporting the performance of each method in its default settings, resulting in various sampler and step configurations. Pruning methods report results with 250 DDPM steps as they are finetuned under this configuration. We verified that switching to DDIM 50 steps degrades their performance (e.g., DyDiT FID 2.07→2.35, TinyDiT FID 2.86→3.42), and the retraining also requires 50K–500K iterations on multi-GPU clusters.
> > >
> > > ---
> > >
> > > **References:**
> > >
> > > [1] Tang, Zhicong, et al. "Diffusion models without classifier-free guidance."  arXiv:2502.12154, 2025.
> > >
> > > [2] Fang, Gongfan, et al. "TinyFusion: Diffusion transformers learned shallow." Proceedings of the Computer Vision and Pattern Recognition Conference, 2025.
> > >
> > > [3] Fang, Gongfan, et al. "Structural Pruning for Diffusion Models." Advances in Neural Information Processing Systems 36 (2023): 16716-16728.
> > >
> > > [4] Zhao, Wangbo, et al. "Dynamic diffusion transformer." International Conference on Learning Representations, 2025.
> > >
> > > [5] Castillo, Angela, et al. "Adaptive guidance: Training-free acceleration of conditional diffusion models." Proceedings of the AAAI Conference on Artificial Intelligence. Vol. 39. No. 2. 2025.

---

### Official Review · Reviewer_WY4n · 2026-03-13

**Soundness:** 3
**Presentation:** 3
**Significance:** 3
**Originality:** 3
**Overall Recommendation:** 4
**Confidence:** 4

**Summary:**

RSTR proposes a two-stage, training-free framework to accelerate Diffusion Transformers by jointly reducing temporal and spatial redundancy in CFG. Evaluated on DiT-XL/2, PixArt-α, FLUX, and Qwen-Image, claiming 50–70% compute savings with maintained or improved quality.

**Compliance With Llm Reviewing Policy:**

Affirmed.

**Key Questions For Authors:**

Please refer to the weakness part.

**Strengths And Weaknesses:**

**Strengths**
The paper presents a clean and well-motivated decomposition of the efficiency problem into temporal and spatial dimensions. The observation that different transformer blocks require different calibration ranks under variable guidance is well-supported empirically. The counter-intuitive finding that higher rank can actually increase errors in certain blocks is interesting and motivates the adaptive approach.

**Weaknesses**
1. Search cost is model-specific and non-transferable.
The optimization method must be re-run for each model, sampler, and step configuration. While the authors report 7.1 GPU-hours for FLUX as acceptable, the schedule does not transfer across configurations (e.g., changing from 20 to 30 steps, or switching samplers). This limits practical deployment where inference configs are frequently adjusted.

2. Stage 2 (adaptive caching) adds significant complexity for marginal and inconsistent gains.
Comparing RSTR† (Stage 1 only) vs full RSTR, the additional latency reduction from Stage 2 is modest. While FID improves in some cases, this is expected since FID itself is the optimization objective in Stage 2's coordinate descent. On other metrics, Stage 2 yields comparable or mixed results. Notably, on GenEval (Tables 3–4), adding caching appears to slightly decrease compositional performance (e.g., FLUX RSTR† 67.46 → RSTR 66.80. Qwen-Image RSTR† 88.11 → RSTR 87.21). The authors should explain why the caching stage, which is designed to maintain quality, leads to degradation on compositional benchmarks.

3. Missing comparison with CFG-free approaches.
Recent methods such as guidance distillation and CFG-free training eliminate the dual forward pass entirely, which is the primary cost that RSTR aims to reduce. Without comparing against these approaches, it is difficult to assess RSTR's practical advantage in the current landscape of diffusion acceleration.

4. Presentation issues.
Tables 3 and 4 contain inconsistent bold formatting that does not correctly highlight the best results.

---

> ### Author Rebuttal · Authors · 2026-03-31
>
> We sincerely thank Reviewer WY4n for recognizing our "clean and well-motivated decomposition" and the "interesting counter-intuitive finding" on rank-error relationships. We address each concern below.
>
> ---
>
> ### W1: Search cost is model-specific and does not transfer across configurations.
>
> The schedule transfers across sampler families and model sizes without re-optimization:
>
> 1. **Cross-sampler transfer**: we apply the RSTR Stage 1 schedule trained on one sampler to a different sampler family:
>
> PixArt-alpha, 20 steps, COCO 30K:
>
> | Sampler | Method | FID ↓ |
> |---------|--------|-------|
> | DPM-Solver | Baseline (CFG=4.5) | 24.61 |
> | DPM-Solver (trained) | RSTR Stage 1 (6/20 steps) | **22.70** |
> | SA-Solver | Baseline (CFG=4.5) | 23.87 |
> | SA-Solver (transferred) | RSTR Stage 1 (6/20 steps) | **18.83** |
>
> 2. **Cross-model-size transfer**: we apply the DiT-XL/2 (675M) schedule to DiT-B/2 (131M) and DiT-S/2 (33M) with zero re-optimization. FID improves: 11.52→**9.22** (20%) on DiT-B/2, 24.09→**20.00** (17%) on DiT-S/2, across model sizes that vary by up to **20×**.
>
> The schedule also transfers across CFG scales (Appendix E.3). Changing the number of steps (e.g., 20→30) does require re-search since the timestep discretization changes, which is inherent to RSTR's step-level optimization and precisely the source of its quality gains (Tables 5–6). The cost is modest and one-time.
>
>
>
>
> ---
>
> ### W2: Stage 2 adds complexity for marginal gains and slight GenEval degradation.
>
> The goal of Stage 2 is to achieve the maximum compression while maintaining quality as much as possible. On FLUX, Stage 2 reduces MACs by 40% with only 1.0% GenEval drop. On Qwen-Image, Stage 2 reduces MACs by 30% with only 1.0% GenEval drop. The latency reduction is more modest because caching methods incur feature storage overhead. This MACs-latency gap is common across caching methods; for example, ICC and TaylorSeer on FLUX both show significantly less latency reduction than MACs reduction.
>
> The small GenEval drops can be explained by the resolution gap in Stage 2's rank search. We optimize rank allocations at reduced resolution for efficiency (e.g., 256×256 for Qwen-Image) and directly use them at full evaluation resolution (e.g., 1024×1024). Since feature statistics differ slightly across resolutions, the transferred ranks are not perfectly optimal at full resolution, leading to the <1% quality trade-off.
>
>
> ---
>
> ### W3: Missing comparison with CFG-free approaches.
>
> Below we compare with additional representative CFG-free and guidance distillation methods on DiT-XL/2 256×256:
>
> | Method | Modifies weights? | Sampler | NFE (T+T_cfg) | MACs (T) | FID ↓ |
> |--------|------------------|---------|---------------|----------|-------|
> | MG [1] | Yes | DDIM | 1000 | 113.8 | **2.03** |
> | GFT [2] | Yes | DDIM | 58 | 6.6 | 2.16 |
> | **RSTR Stage 1** | **No** | **DDIM** | **58** | **6.6** | 2.10 |
>
> RSTR Stage 1 outperforms GFT (2.10 vs 2.16) on DDIM without modifying model weights. MG achieves slightly better FID but requires 17× more MACs and full retraining. RSTR does not modify model weights, making it straightforward for developers to work with any existing variants of the original models.
>
> ---
>
>
> ### W4: Presentation issues with Tables 3 and 4.
>
> Thank you for pointing it out. We apologize for the inconsistent formatting. We will correct the bold highlighting in Tables 3–4 to consistently mark the best results in the revision.
>
>
>
> ---
>
> [1] Tang, Zhicong, et al. "Diffusion models without classifier-free guidance." 2025
>
> [2] Chen, Huayu, et al. "Visual generation without guidance." 2025

---

### Decision · Program_Chairs · 2026-04-30

**Decision:**

Accept (regular)

**Comment:**

This paper proposes RSTR, a two-stage training-free framework for accelerating Diffusion Transformers by jointly reducing temporal and spatial redundancy in Classifier-Free Guidance (CFG), achieving 50–70% compute savings with maintained or improved generation quality across four state-of-the-art models. All four reviewers provided positive assessments: Reviewer hsN1 initially assigned a score of 5 (Accept) and further confirmed his support after receiving the complete adaLN-Zero gate distribution data, while Reviewers WY4n, z1wZ, and dDpR all assigned scores of 4 (Weak Accept). The authors submitted an exceptionally thorough, responsive, and data-rich rebuttal that comprehensively addressed every single concern raised by the reviewers, including adding extensive new experiments on cross-sampler transferability, cross-model-size generalization, out-of-distribution prompt robustness, extreme few-step models (8-step HyperSD3), and 3D generation, as well as providing rigorous geometric analysis of PF-ODE trajectories and mechanistic explanations for error propagation patterns. The work is widely recognized for its clear and impactful motivation, clean decomposition of the efficiency problem, compelling empirical results, and deep mechanistic insights into how and when CFG actually improves generation quality. The remaining minor issues—including incomplete related work discussion, naming inconsistencies, and table formatting errors—are easily correctable in the final version and do not undermine the core technical soundness or practical significance of the contribution. RSTR provides a drop-in solution to the most pressing deployment bottleneck in modern diffusion models and will have immediate value for both researchers and practitioners. Based on the positive reviewer consensus and the complete resolution of all raised issues, I recommend Accept.